# On the Optimality of Activations in Implicit Neural Representations

## Abstract

Implicit neural representations (INRs) have recently surged in popularity as a class of neural networks capable of encoding signals as compact, differentiable entities. To capture high-frequency content, INRs often employ techniques such as Fourier positional encodings or non-traditional activation functions like Gaussian, sinusoid, or wavelets. Despite the impressive results achieved with these activations, there has been limited exploration of their properties within a unified theoretical framework. To address this gap, we conduct a comprehensive analysis of these activations from the perspective of sampling theory. Our investigation reveals that, particularly in the context of shallow INRs, sinc activations—previously unused in conjunction with INRs—are theoretically optimal for signal encoding. Additionally, we establish a connection between dynamical systems and INRs and leverage sampling theory to bridge these two paradigms. Notably, we showcase how the implicit architectural regularization inherent to INRs allows for their application in modeling such systems with minimal need for explicit regularizations.

## 1 Introduction

Recently, the concept of representing signals as Implicit Neural Representations (INRs) has garnered widespread attention across various problem domains (Mildenhall et al., 2021; Li et al., 2023; Büsching et al., 2023; Peng et al., 2021; Strümpler et al., 2022). This surge in popularity can be attributed to the remarkable capability of INRs to encode high-frequency signals as continuous representations. Unlike conventional neural networks, which typically process and convert sparse, high-dimensional signals (such as images, videos, text) into label spaces (e.g., one-hot encodings, segmentation masks, text corpora), INRs specialize in encoding and representing signals by consuming low-dimensional coordinates.

However, a significant challenge in representing signals using neural networks is the presence of spectral bias (Rahaman et al., 2019). Neural networks inherently tend to favor learning functions with lower frequencies, which can hinder their ability to capture high-frequency information. To address this challenge, a common approach involves projecting low-dimensional coordinates into a higher-dimensional space while preserving a specific Lipschitz structure through positional encodings (Zheng et al., 2022; Tancik et al., 2020). Prior research has demonstrated that incorporating positional encodings allows INRs to achieve high-rank representations, enabling them to capture fine details (Zheng et al., 2022). Nevertheless, positional encodings have a critical limitation – they struggle to maintain smooth gradients, which can be problematic for optimization (Ramasinghe & Lucey, 2022; Chng et al., 2022). To overcome this limitation, non-traditional activations, such as sinusoids (Sitzmann et al., 2020), Gaussians (Ramasinghe & Lucey, 2022), and wavelets (Saragadam et al., 2023), have emerged as effective alternatives. These unconventional activations facilitate encoding higher frequencies while preserving smooth gradients, and as shown in prior research (Chng et al., 2022; Saratchandran et al., 2023), they are remarkably stable with respect to various optimization algorithms (except for Sinusoid activations).

Until now, prior research that delved into the analysis of activations in INRs has primarily been tied to the specific activations proposed in their respective studies. For instance, Sitzmann et al. (2020) introduced sinusoidal activations and demonstrated their shift invariance and favorable properties for learning natural signals. Ramasinghe & Lucey (2022) explored activations derived from the exponential family, showcasing their high Lipschitz constants that enable INRs to capture sharp variations. More recently, wavelet-based activations Saragadam et al. (2023) were introduced, high-

lighting their spatial-frequency concentration and suitability for representing images. However, this fragmented approach has obscured the broader picture, making it difficult to draw connections and conduct effective comparisons among these activations. In contrast, our research unveils a unified theory of INR activations through the lens of Nyquist-Shannon sampling. Specifically, we show that, under mild conditions, activations in INRs can be considered as generator functions that facilitate the reconstruction of a given signal from sparse samples. Leveraging this insight, we demonstrate that activations in the form of $\frac{sin(x)}{x}$ (known as the sinc function) theoretically enable shallow INRs to optimally reconstruct a given signal while preserving smooth gradients. Note that, to best of our knowledge, sinc activations have not been used with INRs previously. Furthermore, we validate these insights in practical scenarios using deeper INRs across tasks involving images and neural radiance fields (NeRF).

The inspiring proficiency of sinc-activated INRs in signal reconstruction sparks an intriguing possibility: the effective modeling of more intricate signals through the utilization of these activations. To explore this hypothesis, we turn our attention to chaotic dynamical systems. We observe that both dynamical systems and INRs can be approached from a signal processing perspective. Dynamical systems can be regarded as multi-dimensional signals evolving over time, while modeling such systems from limited measurements of physical quantities resembles the task of reconstructing multi-dimensional signals from discrete samples. INRs, in their own right, share a similar objective, aiming to encode and reconstruct a continuous signal from discrete coordinates and corresponding samples. Drawing inspiration from this connection, we establish parallels between dynamical systems and INRs, leveraging sampling theory to bridge these two paradigms. Ultimately, our research not only demonstrates the superior performance of sinc-activated INRs in modeling dynamical systems but also provides a theoretical explanation for this advantage.

## 2 A SAMPLING PERSPECTIVE ON INRS

In this section, we present our primary theoretical insights, showcasing how sampling theory offers a fresh perspective on understanding the optimality of activations in INRs.

### 2.1 IMPLICIT NEURAL REPRESENTATIONS

We consider INRs of the following form: Consider an $L$-layer network, $F_L$, with widths $\{n_0, \ldots, n_L\}$. The output at layer $l$, denoted $f_l$, is given by

$$f_l(x) = \begin{cases} x, & \text{if } l = 0 \\ \phi(W_l F_{l-1} + b_l), & \text{if } l \in [1, \ldots, L-1] \\ W_{L-1} F_{L-1} + b_L, & \text{if } l = L \end{cases} \tag{1}$$

where $W_l \in \mathbb{R}^{n_l \times n_{l-1}}$, $b_l \in \mathbb{R}^{n_l}$ are the weights and biases respectively of the network, and $\phi$ is a non-linear activation function.

### 2.2 CLASSICAL SAMPLING THOERY

Sampling theory considers bandlimited signals, which are characterized by a limited frequency range. Formally, for a continuous signal denoted as $f$, being bandlimited to a maximum frequency of $\Omega$ implies that its Fourier transform, represented as $\widehat{f}(s)$, equals zero for all $|s|$ values greater than $\Omega$. If we have an $\Omega$-bandlimited signal $f$ belonging to the space $L^2(\mathbb{R})$, then the Nyquist-Shannon sampling theorem (as referenced in Zayed (2018)) provides a way to represent this signal as $f(x) = \sum_{n=-\infty}^{\infty} f\left(\frac{n}{2\Omega}\right) \text{sinc}\left(2\Omega\left(x - \frac{n}{2\Omega}\right)\right)$, where the equality means converges in the $L^2$ sense. Essentially, by sampling the signal at regularly spaced points defined by $\frac{n}{2\Omega}$ for all integer values of $n$, and using shifted sinc functions, we can reconstruct the original signal. However, this requires us to sample at a rate of at least $2\Omega$-Hertz.

In theory, perfect reconstruction necessitates an infinite number of samples, which is impractical in real-world scenarios. It's crucial to acknowledge that the sampling theorem is an idealization, not universally applicable to real signals due to their non-bandlimited nature. However, as natural signals often exhibit dominant frequency components at lower energies, we can effectively approximate the original signal by projecting it into a finite-dimensional space of bandlimited functions, enabling robust reconstruction.

## 2.3 Optimal Activations via Riesz Sampling

In the previous section, we discussed how an exact reconstruction of a bandlimited signal could be achieved via a linear combination of shifted $\mathrm{sinc}$ functions. Thus, it is intriguing to explore if an analogous connection can be drawn to coordinate networks. We will take a general approach and consider spaces of the form

$$V(F) = \left\{ s(x) = \sum_{k \in \mathbb{Z}} a(k) F(x-k) : a \in l^2(\mathbb{R}) \right\}, \tag{2}$$

where $l^2(\mathbb{R})$ denotes the Hilbert space of square summable sequences over the integers $\mathbb{Z}$. The space $V(F)$ should be seen as a generalisation of the space of bandlimited functions occurring in the Shannon sampling theorem.

**Definition 2.1.** *The family of translates $\{F_k = F(x-k)\}_{k \in \mathbb{Z}}$ is a Riesz basis for $V(F)$ if the following two conditions hold: 1) $A||a||_{l^2}^2 \le \left|\left| \sum_{k \in \mathbb{Z}} a(k) F_k \right|\right|^2 \le B||a||_{l^2}^2$ , $\forall a(k) \in l^2(\mathbb{R})$. 2) $\sum_{k \in \mathbb{Z}} F(x+k) = 1$, $\forall x \in \mathbb{R}$ (PUC).*

Observe that if $s = \sum_k a(k) F_k = 0$ then the lower inequality in 1. implies $a(k) = 0$ for all $k$. In other words, the basis functions $F_k$ are linearly independent, which in turn implies each signal $s \in V(F)$ is uniquely determined by its coefficient sequence $a(k) \in l^2(\mathbb{R})$. The upper inequality in 1. implies that the $L^2$ norm of a signal $s \in V(F)$ is finite, implying that $V(F)$ is a subspace of $L^2(\mathbb{R})$.

The second condition is known as the *partition of unity condition* (**PUC**). It allows the capability of approximating a signal $s \in V(F)$ as closely as possible by selecting a sample step that is sufficiently small. This can be seen as a generalisation of the Nyquist criterion, where in order to reconstruct a band-limited signal, a sampling step of less than $\frac{\pi}{2\omega_{\max}}$ must be chosen, where $\omega_{\max}$ is the highest frequency present in the signal $s$.

**Definition 2.2.** *The family of translates $\{F_k = F(x-k)\}_{k \in \mathbb{Z}}$ is a weak Riesz basis for $V(F)$ if only condition 1. from defn. 2.1 holds.*

The following proposition considers activations in INRs and the $\mathrm{sinc}$ function. Specifically, we show that $\mathrm{sinc}$ forms a Riesz basis, Gaussian and wavelets form weak Reisz bases, and ReLU and Sinusoid does not form Riesz/weak Riesz bases. The proof is given in appendix A.1.

**Proposition 2.3.**    *1. Let $F(x) = \mathrm{sinc}(x) = \frac{\sin(x)}{x}$ (defined to be 1 at $x = 0$), the family $\{\mathrm{sinc}(x-k)\}_{k \in \mathbb{Z}}$ forms a Riesz basis where $V(\mathrm{sinc})$ is the space of signals with frequency bandlimited to $[-1/2, 1/2]$.*

2. *Let $F(x) = G_s(x) := e^{-x^2/s^2}$, for some fixed $s > 0$, the family $\{G_s(x-k)\}_{k \in \mathbb{Z}}$ forms a weak Riesz basis for the space $V(G_s)$ but not a Riesz basis. In this case $V(G_s)$ can be interpreted as signals whose Fourier transform has Gaussian decay, where the rate of decay will depend on $s$.*

3. *Let $F(x) = \Psi(x)$ denote a wavelet. In general wavelets form a weak Riesz basis but not all form a Riesz basis.*

4. *Let $F(x) = ReLU(x)$, the family $\{ReLU(x-k)\}_{k \in \mathbb{Z}}$ does not form a Riesz/weak Riesz basis as it violates condition 1. from defn. 2.1.*

5. *Let $F(x) = \sin(\omega x)$, for $\omega$ a fixed frequency parameter, the family $\{sin(\omega(x-k))\}_{k \in \mathbb{Z}}$ does not form a Riesz/weak Riesz basis as it violates condition 1. from defn. 2.1.*

Interestingly, to fit INRs into the above picture, observe that the elements in $V(F)$ that are finite sums can be represented by INRs with activation $F$ (this is explicitly proved in the next theorem, see also appendix A.1). Thus, it follows that signals in $V(F)$ that have an infinite number of non-zero summands can be approximated by INRs, the proof can be found in app. A.2.1.

**Theorem 2.4.** *Suppose the family of functions $\{F(x-k)\}_{k \in \mathbb{Z}}$ forms a weak Riesz basis for the space $V(F)$. Let $g$ be a signal in $V(F)$ and let $\epsilon > 0$ be given. Then there exists a 2-layer INR $f$, with a parameter set $\theta$, $F$ as the activation, and $n(\epsilon)$ neurons in the hidden layer, such that*

$$||f(\theta) - g||_{L^2} < \epsilon.$$

The primary limitation of Theorem 2.4 lies in its applicability solely to signals within the domain of $V(F)$. This prompts us to inquire whether Riesz bases can be employed to approximate arbitrary $L^2$-functions, even those outside the confines of $V(F)$. The significance of posing this question lies in the potential revelation that, if affirmed, INRs can also approximate such signals. This would, in turn, demonstrate the universality of $F$-activated INRs within the space of $L^2(\mathbb{R})$ functions.

We now show, in order to be able to approximate arbitrary signals in $L^2(\mathbb{R})$ the partition of unity condition, condition 2 of defn. 2.1, plays a key role. To analyse this situation, we introduce the scaled signal spaces.

**Definition 2.5.** *For a fixed $\Omega > 0$, let $V_\Omega(F) = \left\{ s_\Omega = \sum_{k \in \mathbb{Z}} a_\Omega(k) F(\frac{x}{\Omega} - k) : a \in l^2(\mathbb{R}) \right\}$. We call $V_\Omega(F)$ an $\Omega$-scaled signal space.*

**Example 2.6.** *The canonical example of an $\Omega$-scaled signal space is given by taking $F(x) = \text{sinc}(2\Omega x)$. In this case $V_\Omega(F)$ is the space of $\Omega$-bandlimited signals.*

The difference between $V_\Omega(F)$ and $V(F)$ is that in the former the basis functions are scaled by $\Omega$. Previously, we remarked that one of the issues in applying thm. 2.4 is that it does not provide a means for approximating general signals in $L^2(\mathbb{R})$ by INRs. What we wish to establish now is that given an arbitrary signal $s \in L^2(\mathbb{R})$ and an approximation error $\epsilon > 0$, if $\{F(x - k)\}_{k \in \mathbb{Z}}$ is a Riesz basis then there exists a scale $\Omega(\epsilon)$, that depends on $\epsilon$, such that the scaled signal space $V_{\Omega(\epsilon)}(F)$ can approximate $s$ to within $\epsilon$ in the $L^2$-norm. We will follow the approach taken by Unser (2000) and give a brief overview of how to proceed. More details can be found in app. A.

In order to understand how we can reconstruct a signal to within a given error using $V_\Omega(F)$, we define the approximation operator $A_\Omega : L^2(\mathbb{R}) \to V_\Omega(F)$ by

$$A_\Omega(s(x)) = \sum_{k \in \mathbb{Z}} \left( \int_{\mathbb{R}} s(y) \widetilde{F}(\frac{y}{\Omega} - k) \frac{dy}{\Omega} \right) F(\frac{x}{\Omega} - k) \tag{3}$$

where $\widetilde{F}$ is a suitable analysis function from a fixed test space. We will not go into the details of how to construct $\widetilde{F}$ but for now will simply assume such a $\widetilde{F}$ exists and remark that its definition depends on $F$. For details on how to construct $\widetilde{F}$ we refer the reader to app. A.1.1. The quantity $\int s(y) \widetilde{F}(\frac{y}{\Omega} - k) \frac{dy}{\Omega}$ is to be thought of as the coefficients $a_\Omega(k)$ in reconstructing $s$.

The approximation error is defined as

$$\epsilon_s(\Omega) = ||s - A_\Omega(s)||_{L^2}. \tag{4}$$

The goal is to understand how we can make the approximation error as small as we like by choosing $\Omega$ and the right analysis function $\widetilde{F}$.

The general approach to this problem via sampling theory, see Unser (2000) for details, is to go via the average approximation error:

$$\bar{\epsilon}_s(\Omega)^2 = \frac{1}{\Omega} \int_0^\Omega ||s(\cdot - \tau) - A_\Omega(s(\cdot - \tau))||_{L^2}^2 d\tau. \tag{5}$$

Using Fourier analysis, see (Blu & Unser, 1999), it can be shown that $\bar{\epsilon}_s(\Omega)^2 = \int_{-\infty}^{+\infty} E_{\widetilde{F},F}(\Omega\xi) |\hat{s}(\xi)|^2 \frac{d\xi}{2\pi}$ where $\hat{s}$ denotes the Fourier transform of $s$ and $E_{\widetilde{F},F}$ is the error kernel defined by

$$E_{\widetilde{F},F}(\omega) = |1 - \hat{\widetilde{F}}(\omega)\hat{F}(\omega)|^2 + |\hat{\widetilde{F}}(\omega)|^2 \sum_{k \neq 0} |\hat{F}(\omega + 2\pi k)|^2 \tag{6}$$

where $\hat{F}$ and $\hat{\widetilde{F}}$ denote the Fourier transforms of $F$ and $\widetilde{F}$ respectively. Understanding the approximation properties of the shifted basis functions $F_k = F(x - k)$ comes down to analysing the error kernel $E_{\widetilde{F},F}$. The reason being is that the average error $\bar{\epsilon}_s(\Omega)^2$ is a good predictor of the true error $\epsilon_s(\Omega)^2$ as the following theorem, from Blu & Unser (1999), shows.

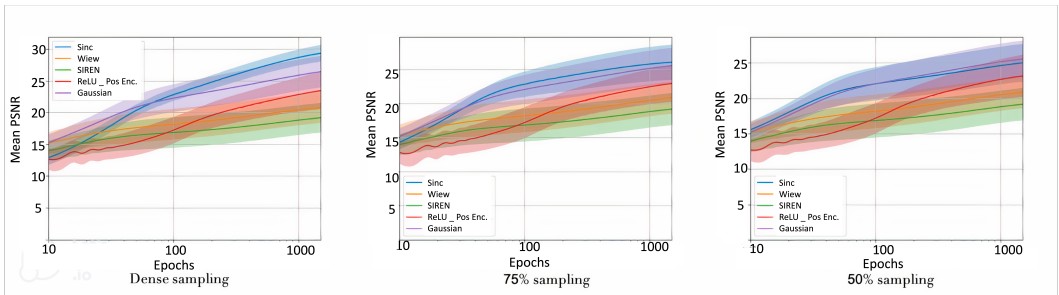

Figure 1: **Comparison of Image reconstruction across different INRs over DIVK dataset.** We run a grid search to find the optimal parameters for each INR. Note that a single optimal parameter setting is used for each activation, across all the images in the dataset.

**Theorem 2.7.** *The $L^2$ approximation error $\epsilon_s(\Omega)^2$ can be written as*

$$\epsilon_s(\Omega)^2 = \left( \int_{-\infty}^{\infty} E_{\widetilde{F},F}(\Omega\xi)|\hat{s}(\xi)|^2 \frac{d\xi}{2\pi} \right)^{1/2} + \epsilon_{corr} \tag{7}$$

*where $\epsilon_{corr}$ is a correction term negligible under most circumstances. Specifically, if $f \in W_2^r$ (Sobolev space of order $r$, see appendix A) with $r > \frac{1}{2}$, then $|\epsilon_{corr}| \leq \gamma\Omega^r||s^{(r)}||_{L^2}$ where $\gamma$ is a known constant and Moreover, $|\epsilon_{corr}| = 0$ provided the signal $s$ is band-limited to $\frac{\pi}{\Omega}$.*

Thm. 2.7 shows that the dominant part of the approximation error $\epsilon_s(\Omega)$ is controlled by the average error $\bar{\epsilon}_s(\Omega)$. This means that in order to show that there exists a scale $\Omega$ such that the scaled signal space $V_\Omega(F)$ can be used to approximate $s \in L^2(\mathbb{R})$ up to any given error, it suffices to show that

$$\lim_{\Omega \to 0} E_{\widetilde{F},F} \to 0. \tag{8}$$

The following lemma gives the required condition to guarantee vanishing of the error kernel as $\Omega \to 0$.

**Lemma 2.8.** *If the family of shifted basis function $\{F(x - k)\}_{k \in \mathbb{Z}}$ satisfies the condition*

$$\sum_{k \in Z} F(x + k) = 1, \forall x \in \mathbb{R} \tag{9}$$

*then $\lim_{\Omega \to 0} E_{\widetilde{F},F} \to 0$, for any $\widetilde{F} \in \overline{\mathcal{S}}$, where $\overline{\mathcal{S}}$ is the space of Schwartz functions $f$ whose Fourier transform satisfies $\hat{f}(0) = 1$.*

The above lemma shows the importance of the partition of unity condition (**PUC**), condition 2, from defn. 2.1. A sketch of the proof of lem. 2.8 is given in app. A.1.1 together with details on the space $\overline{\mathcal{S}}$ and its relevance to the error kernel $E_{\widetilde{F},F}$.

From prop. 2.3, we see that the sinc function has vanishing error kernel as the scale $\Omega \to 0$. However, a Gaussian does not necessarily have vanishing error kernel as $\Omega \to 0$.

Lem. 2.8 immediately implies the following approximation result, proof can be found in app. A.1.1

**Proposition 2.9.** *Let $s \in L^2(\mathbb{R})$ and $\epsilon > 0$. Assume the shifted functions $\{F(x - k)\}_{k \in \mathbb{Z}}$ form a Riesz basis for $V(F)$. Then there exists an $\Omega > 0$ and an $f_\Omega$ in $V_\Omega(F)$ such that*

$$||s - f_\Omega||_{L^2} < \epsilon. \tag{10}$$

Prop. 2.9 implies that the signal $s$ can be approximated by basis functions given by shifts of $F$ with bandwidth $1/\Omega$.

Using prop. 2.9 we obtain a universal approximation result for neural networks employing Riesz bases as their activation functions.

**Theorem 2.10.** *Let $s \in L^2(\mathbb{R})$ and $\epsilon > 0$. Assume the shifted functions $\{F(x-k)\}_{k \in \mathbb{Z}}$ form a Riesz basis for $V(F)$. Then there exists a 2-layer INR $\mathcal{N}$, with a parameter set $\theta$, activation $F$, $n(\epsilon)$ neurons in the hidden layer, and an $\Omega > 0$ such that*

$$||\mathcal{N}(\theta) - s||_{L^2} < \epsilon$$

*where $\mathcal{N}(\theta)$ employs $F_\Omega$ as its activation in the hidden layer, where $F_\Omega(x) = F(\frac{1}{\Omega}x)$.*

**Remark 2.11.** *Thm. 2.10 shows why* sinc *being able to generate a Riesz basis is an optimal condition to satisfy for an activation function. Note that in general, any function in $L^2(\mathbb{R})$ that generates a Riesz basis will be optimal in this sense. Furthermore, out of the activations that practitioners in the ML community use, such as* sinc, sin, *Gaussian,* tanh, ReLU, sigmoid, *we find* sinc *is the optimal.*

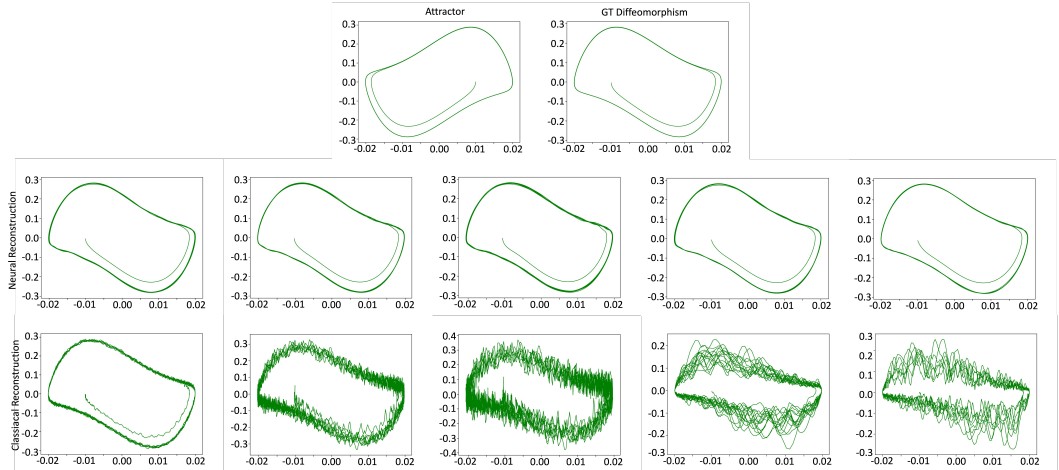

Figure 2: Discovering the dynamics from partial observations. We use the Vanderpol system for this illustration. *Top row:* the original attractor and the diffeomorphism obtained by the SVD decomposition of the Hankel matrix (see Sec. 3.3.1) without noise. *Third row:* The same procedure is used to obtain the reconstructions with noisy, random, and sparse samples. *Second row:* First, a sinc-INR is used to obtain a continuous reconstruction of the signal from discrete samples, which is then used as a surrogate signal to resample measurements. Afterwards, the diffeomorphisms are obtained using those measurements. As shown, sinc-INRs are able to recover the dynamics more robustly with noisy, sparse, and random samples.

Theorem 2.10 underscores the significance of the activation function within an INR when it comes to signal reconstruction in the $L^2(\mathbb{R})$. Specifically, as exemplified in prop 2.3, it becomes evident that an INR equipped with a sinc activation function can achieve reconstructions of signals in $L^2(\mathbb{R})$ up to any accuracy, rendering it the optimal choice for the INR architecture. See App. C for the connection of the above analysis to the universal approximation theorem.

## 3 EXPERIMENTS

In this section, we aim to compare the performance of different INR activations. First, we focus on image and NeRF reconstructions and later move on to dynamical systems.

### 3.1 IMAGE RECONSTRUCTION

A critical problem entailed with INRs is that they are sensitive to the hyperparameters in activation functions (Ramasinghe & Lucey, 2022). That is, one has to tune the hyperparameters of the activations to match the spectral properties of the encoded signal. Here, we focus on the robustness of activation parameters when encoding different signals. To this end, we do a grid search and find the *single* best performing hyperparameter setting for *all* the images in a sub-sampled set of the DIV2K dataset (Agustsson & Timofte, 2017) released by Tancik et al. (2020). For example, for sinc activations, we experiment with different bandwidth parameters, each time fixing it across the entire dataset. Then, we select the bandwidth parameter that produced the best results. Since all

| Activation | Rossler | | | Lorenz | | |
|---|---|---|---|---|---|---|
| | $n = 0.1$ | $n = 0.5$ | $n = 1.$ | $n = 0.1$ | $n = 0.5$ | $n = 1.$ |
| Baseline | 42.1 | 29.8 | 22.3 | 43.8 | 28.2 | 20.3 |
| Gaussian | 46.6 | 37.1 | 33.6 | 45.7 | 39.1 | 35.7 |
| Sinusoid | 45.1 | 36.6 | 32.1 | 42.1 | 37.4 | 30.3 |
| Wavelet | 40.3 | 35.9 | 30.9 | 38.2 | 37.3 | 31.8 |
| Sinc | **48.9** | **42.8** | **38.5** | **46.2** | **40.9** | **39.3** |

Table 2: SINDy reconstructions (PSNR) with different noise levels ($n =$ standard deviation of the Gaussian noise) injected into samples.

the compared activations contain a tunable parameter, we perform the same for all the activations and find the best parameters for each. This dataset contains images of varying spectral properties: 32 images of *Text* and *Natural scenes*, each. We train with different sampling rates and test against the full ground truth image. The PSNR plots are shown in Fig. 1. As depicted, sinc activation performs better or on-par with other activations. We use 4-layer networks with 256 width for these experiments.

## 3.2 NEURAL RADIANCE FIELDS

NeRFs are one of the key applications of INRs, popularized by Mildenhall et al. (2021). Thus, we evaluate the performance of sinc-INRs in this setting. Table. 1 demonstrates quantitative results. We observed that all the activations perform on-par with NeRF reconstructions with proper hyperparameter tuning, where sinc outperformed the rest marginally.

## 3.3 DYNAMICAL SYSTEMS

It is intriguing to see if the superior signal encoding properties of sinc-INRs (as predicted by the theory) would translate to a clear advantage in a challenging setting. To this end, we choose dynamical (chaotic) systems as a test bed.

Dynamical systems can be defined in terms of a time dependant state space $\mathbf{x}(t) \in \mathbb{R}^D$ where the time evolution of $\mathbf{x}(t)$ can be described via a differential equation,

| Activation | PSNR | SSIM |
|---|---|---|
| Gaussian | 31.13 | 0.947 |
| Sinusoid | 28.96 | 0.933 |
| Wavelet | 30.33 | 0.941 |
| Sinc | **31.37** | **0.947** |

Table 1: **Quantitative comparison in novel view synthesis on the real synthetic dataset (Mildenhall et al., 2021).** sinc-INRs perform on-par with other activations.

$$\frac{d\mathbf{x}(t)}{dt} = f(\mathbf{x}(t), \alpha), \qquad (11)$$

where $f$ is a non-linear function and $\alpha$ are a set of system parameters. The solution to the differential equation 11 gives the time dynamics of the state space $\mathbf{x}(t)$. In practice, we only have access to discrete measurements $[\mathbf{y}(t_1), \mathbf{y}(t_2), \dots \mathbf{y}(t_Q)]$ where $\mathbf{y}(t) = g(\mathbf{x}(t)) + \eta$ and $\{t_n\}_{n=1}^{Q}$ are discrete instances in time. Here, $g(\cdot)$ can be the identity or any other non-linear function, and $\eta$ is noise. Thus, the central challenge in modeling dynamical systems can be considered as recovering the characteristics of the state space from such discrete observations.

We note that modeling dynamical systems and encoding signals using INRs are analogous tasks. That is, modeling dynamical systems can be interpreted as recovering characteristics of a particular system via measured physical quantities over time intervals. Similarly, coordinate networks are used to recover a signal given discrete samples.

### 3.3.1 DISCOVERING THE DYNAMICS OF LATENT VARIABLES

In practical scenarios, we often encounter limitations in measuring all the variables influencing a system's dynamics. When only partial measurements are available, deriving a closed-form model for the system becomes challenging. However, Takens' Theorem (refer to App. B) offers a significant insight. It suggests that under certain conditions, augmenting partial measurements with delay embeddings can produce an attractor diffeomorphic to the original one. This approach is remarkably powerful, allowing for the discovery of complex system dynamics from a limited set of variables.

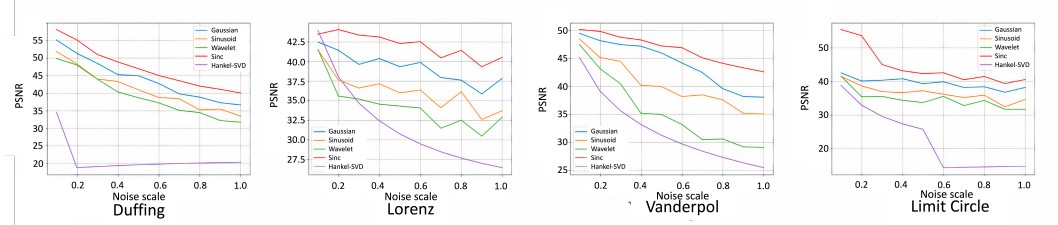

Figure 3: Quantitative comparison on discovering the dynamics of latent variables using INRs vs classical methods.

**Time Delay Embedding.** To implement this, we start with discrete time samples of an observable variable $y(t)$. We construct a Hankel matrix $\mathbf{H}$ by augmenting these samples as *delay embeddings* in each row:

$$\mathbf{H} = \begin{bmatrix} y_1(t_1) & y_1(t_2) & \ldots & y_1(t_n) \\ y_1(t_2) & y_1(t_3) & \ldots & y_1(t_{n+1}) \\ \vdots & \vdots & \ddots & \vdots \\ y_1(t_m) & y_1(t_{m+1}) & \ldots & y_1(t_{m+n+1}) \end{bmatrix}. \tag{12}$$

According to Takens' Theorem, the dominant eigenvectors of this Hankel matrix encapsulate dynamics that are diffeomorphic to the original attractor. For our experiment, we utilize systems such as the Vanderpol, Limit cycle attractor, Lorenz, and Duffing equations (see D). We generate 5000 samples, spanning from 0 to 100, to form the Hankel matrix. Subsequently, we extract its eigenvectors and plot them to visualize the surrogate attractor that mirrors the original attractor. To assess the method's robustness against noise, we introduce noise into the $y(t)$ samples from a uniform distribution $\eta \sim U(-n, n)$, varying $n$.

To demonstrate the efficacy of Implicit Neural Representations (INRs) in this context, we employ a sinc-INR to encode the original measurements as a continuous signal. Initially, we train a sinc-INR using discrete pairs of $t$ and $y(t)$ as inputs and labels. Then, we use the sampled values from the INR as a surrogate signal to create the Hankel matrix, which yields robust results. Interestingly, the continuous reconstruction from the sinc-INR requires sparser samples (with $n\tau = 0.2$)), thus overcoming a restrictive condition typically encountered in this methodology.

The results, as depicted in Fig.2 and Fig.3, clearly demonstrate that sinc-INRs can accurately recover the dynamics of a system from partial, noisy, random, and sparse observations. In contrast, the performance of classical methods deteriorates under these conditions, underscoring the advantage of the sinc-INR approach in handling incomplete and imperfect data.

## 4 Discovering governing equations

The SINDy algorithm is designed to deduce the governing equations of a dynamical system from discrete observations of its variables. Consider observing the time dynamics of a $D$-dimensional variable $\mathbf{y}(t) = [y_1(t), \ldots, y_d(t)]$. For observations at $N$ time stamps, we construct the matrix $\mathbf{Y} = [\mathbf{y}(t_1), \mathbf{y}(t_2), \ldots \mathbf{y}(t_N)]^T \in \mathbb{R}^{N \times D}$. The initial step in SINDy involves computing $\dot{\mathbf{Y}} = [\dot{\mathbf{y}}(t_1), \dot{\mathbf{y}}(t_2), \ldots \dot{\mathbf{y}}(t_N)]^T \in \mathbb{R}^{N \times D}$, achieved either through finite difference or continuous approximation techniques. Subsequently, an augmented library $\Theta(\mathbf{Y})$ is constructed, composed of predefined candidate nonlinear functions of $\mathbf{Y}$'s columns, encompassing constants, polynomials, and trigonometric terms, as illustrated below:

$$\Theta(\mathbf{Y}) = \begin{bmatrix} y_1^2(t_1) & y_2^2(t_1) & \ldots & \sin(y_1(t_1))\cos(y_2(t_1)) & \ldots & y_d(t_1)y_2^2(t_1) \\ y_1^2(t_2) & y_2^2(t_2) & \ldots & \sin(y_1(t_2))\cos(y_2(t_2)) & \ldots & y_d(t_2)y_2^2(t_2) \\ \vdots & \vdots & \ldots & \vdots & \ldots & \vdots \\ y_1^2(t_N) & y_2^2(t_N) & \ldots & \sin(y_1(t_N))\cos(y_2(t_N)) & \ldots & y_d(t_N)y_2^2(t_N) \end{bmatrix}$$

SINDy then seeks to minimize the loss function:

$$L_S = ||\dot{\mathbf{Y}} - \Theta(\mathbf{Y})\Gamma||_2^2 + \lambda||\Gamma||_1^2, \tag{13}$$

where $\Gamma$ is a sparsity matrix initialized randomly that enforces sparsity.

INRs introduce two significant architectural biases here. When we train an INR using $\{t_n\}_{n=1}^N$ and $\{\mathbf{y}(t_n)\}_{n=1}^N$ as inputs and labels, it allows us to reconstruct a continuous representation of $\mathbf{y}(t)$. By controlling the frequency parameter $\omega$ of sinc functions during training, we can filter out high-frequency noise in $\mathbf{y}$. Additionally, $\dot{\mathbf{y}}$ measurements can be obtained by calculating the Jacobian of the network, taking advantage of the smooth derivatives of sinc-INRs. We then replace $\dot{\mathbf{Y}}$ and $\mathbf{Y}$ in Eq. 13 with values obtained from the INR, keeping the rest of the SINDY algorithm unchanged.

For our experiment, we employ the Lorenz and Rossler systems (refer to D), generating 1000 samples from 0 to 100 at intervals of 0.1 to create $\mathbf{Y}$. We introduce noise from a uniform distribution $\eta \sim U(-n, n)$, varying $n$. As a baseline, we compute $\dot{\mathbf{Y}}$ using spectral derivatives, a common method in numerical analysis and signal processing for computing derivatives through spectral methods. This involves translating the function's derivative in the time or space domain to a multiplication by $i\omega$ in the frequency domain. The reason for choosing spectral derivatives is empirical; After evaluating various methods to compute $\dot{\mathbf{Y}}$, including finite difference methods and polynomial approximations, we empirically selected spectral derivatives for the best baseline. As a competing method, for each noise scale, we use a sinc-INR to compute both $\dot{\mathbf{Y}}$ and $\mathbf{Y}$ as described. Utilizing the SINDy algorithm for both scenarios, we obtain the governing equations for each system. The dynamics recovered from these equations are compared in Fig. 7 (Appendix) and Table 2. Remarkably, the sinc-INR approach demonstrates robust results at each noise level, surpassing the baseline. For this experiment, we use 4-layer INRs with each layer having a width of 256.

## 5 RELATED WORK

**INRs.** INRs, pioneered by Mildenhall et al. (2021), have gained prominence as an effective architecture for signal reconstruction. Traditionally, such architectures employed activations such as ReLU and Sigmoid. However, these activations suffer from spectral bias, limiting their effectiveness in capturing high-frequency content (Rahaman et al., 2019). To overcome this limitation, Mildenhall et al. (2021) introduced a positional embedding layer to enhance high-frequency modeling. Meanwhile, Sitzmann et al. (2020) proposed SIREN, a sinusoidal activation that eliminates the need for positional embeddings but exhibits instability with random initializations. In contrast, Ramasinghe & Lucey (2022) introduced Gaussian-activated INRs, showcasing robustness to various initialization schemes. More recently, wavelet activations were proposed by Saragadam et al. (2023) with impressive performance. Yet, the theoretical optimality of these activation functions in the context of signal reconstruction has largely eluded investigation. In this study, we aim to address this gap by examining the selection of activation functions through the lens of sampling theory.

**Data driven dynamical systems modeling.** Numerous approaches have been explored for the data-driven discovery of dynamical systems, employing various techniques. These methodologies include nonlinear regression (Voss et al., 1999), empirical dynamical modeling (Ye et al., 2015), normal form methods (Majda et al., 2009), spectral analysis (Giannakis & Majda, 2012), Dynamic Mode Decomposition (DMD) (Schmid, 2010; Kutz et al., 2016), as well as compressed sensing and sparse regression within a library of candidate models (Reinbold et al., 2021; Wang et al., 2011; Naik & Cochran, 2012; Brunton et al., 2016). Additionally, reduced modeling techniques like Proper Orthogonal Decomposition (POD) (Holmes et al., 2012; Kirby, 2001; Sirovich, 1987; Lumley, 1967), both local and global POD methods (Schmit & Glauser, 2004; Sahyoun & Djouadi, 2013), and adaptive POD methods (Singer & Green, 2009; Peherstorfer & Willcox, 2015) have been widely applied in dynamical system analysis. Koopman operator theory in conjunction with DMD methods has also been utilized for system identification (Budišić et al., 2012; Mezić, 2013).

## 6 CONCLUSION

In this work, we offer a fresh view-point on INRs using sampling theory. In this vein, we show that sinc activations are optimal for encoding signals in the context of shallow networks. We conduct experiments with image reconstructions, NeRFs and dynamical systems to showcase that these theoretical predictions hold at a practical level with deeper networks.

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

## A  Proofs of results in section 2.3

### A.1  Preliminaries

We recall the definition of the space of square integrable functions on $\mathbb{R}$, which we denote by $L^2(\mathbb{R})$, and is defined as the vector space of equivalence classes of Lebesgue measurable functions on $\mathbb{R}$ with the following inner product

$$\langle f, g \rangle_{L^2} = \int_{\mathbb{R}} f \cdot g. \tag{14}$$

We will also need to make use of the Sobolev spaces of order $r$, denoted by $W_2^r(\mathbb{R})$. We define this space as the space of $L^2$-functions that have $r$ weak derivatives that are also in $L^2(\mathbb{R})$.

***Proof of prop. 2.3.*** The function $\mathrm{sinc}(x)$ is in $L^2(\mathbb{R})$ and furthermore the translates $\mathrm{sinc}(x-k)$, for $k \in \mathbb{Z}$, form an orthonormal basis of $L^2(\mathbb{R})$. Hence $\mathrm{sinc}(x)$ satisfies the first condition of a Riesz basis with $A = B = 1$.

The next step is to check that the partition of unity condition holds. In order to do this we will make use of the Poisson summation formula (Stein & Shakarchi, 2011) that states that for a function $f \in L^2(\mathbb{R})$ we have

$$\sum_{k \in \mathbb{Z}} f(x+k) = \sum_{n \in \mathbb{Z}} \hat{f}(2\pi n) e^{2\pi i n x}. \tag{15}$$

Using the Poisson summation formula, we can rewrite the partition of unity condition, see cond. 2 in defn. 2.1, as

$$\sum_{n \in \mathbb{Z}} \hat{f}(2\pi n) e^{2\pi i n x} = `1. \tag{16}$$

We then observe that the Fourier transform of $\mathrm{sinc}(x)$ is given by the characteristic function $\chi_{[-1,1]}$ on the set $[-1, 1]$. I.e. $\chi_{[-1,1]}$ takes the value 1 on $[-1, 1]$ and 0 elsewhere (Stein & Shakarchi, 2011). The proof now follows by observing that $\chi_{[-1,1]}(2\pi n) = 1$ for $n = 0$ and 0 for $n \neq 0$. We then see that equation 16 is true for $\mathrm{sinc}(x)$ and thus $\mathrm{sinc}(x)$ forms a Riesz basis.

The proof that the Gaussian $\phi = e^{-x^2/2s^2}$ does not form a Riesz basis and only a weak Riesz basis follows the same strategy as above. The first step is to note that translates of the Gaussian: $\phi_k = e^{-(x-k)^2/2s^2}$ all lie in $L^2(\mathbb{R})$ for any $k \in \mathbb{Z}$. This establishes the upper bound in condition 1 of the Riesz basis definition. To prove the lower bound in condition 1, we use an equivalent definition of condition 1 in the Fourier domain given by

$$A \leq \sum_{k \in \mathbb{Z}} |\hat{\phi}(\xi + 2k\pi)|^2 \leq B \tag{17}$$

where $\hat{\phi}$ denotes the Fourier transform of $\phi$ and $\xi$ the frequency variable in the Fourier domain. The equivalence of equation 17 with the Riesz basis definition given in defn. 2.1 follows by noting that defn. 2.1 is translation invariant, see Aldroubi et al. (1994) for explicit details. We then observe that in the case of a Gaussian the term $\hat{\phi}(\xi + 2k\pi)$ is given by $e^{-(\xi+2k\pi)^2 s^2/2}$, which follows from the fact that the Fourier transform of a Gaussian is another Gaussian, see Stein & Shakarchi (2011). The final observation to make is that the sum

$$\sum_{k \in \mathbb{Z}} |\hat{\phi}(\xi + 2k\pi)|^2 \geq |\hat{\phi}(\xi)|^2 \tag{18}$$

for any $\xi \in \mathbb{R}$ and that we only need to consider $\xi \in [0, 2\pi]$ from the symmetry of the Gaussian about the y-axis and the fact that for any $\xi$ outside of $[0, 2\pi]$, there exists some $k \in \mathbb{Z}$ such that the translate $\xi + 2k\pi$ lies in $[0, 2\pi]$. The lower bound in equation 17 then follows by taking $0 < A \leq e^{-(2\pi s)^2}$.

In order to show that the Gaussian $\phi$ does not satisfy the partition of unity condition. We go through the formulation equation 16. In this case this formula reads

$$\sum_{n \in \mathbb{Z}} e^{-(2\pi n)^2 s^2/2} e^{2\pi i n x} = 1. \tag{19}$$

We now observe it is impossible for this equality to hold due to the gaussian decay of the function $e^{-(2\pi n)^2 s^2/2}$. In particular for $x = 0$ the condition becomes

$$\sum_{n \in \mathbb{Z}} e^{-(2\pi n)^2 s^2/2} = 1. \tag{20}$$

The left hand side is clearly greater than 1, and thus we see that the condition cannot hold. This proves that a Gaussian $e^{-x^2/2s^2}$ can only define a weak Riesz basis.

In general, the Fourier transform of a wavelet is localized in phase and frequency, hence as in the case of the Gaussian above, they will be in $L^2(\mathbb{R})$ and form a weak Riesz basis but in general they might not form a Riesz basis. Conditions have been given for a wavelet to form a Riesz basis, see Sun & Zhou (2002), though this is outside the scope of this work.

In order to form a Riesz basis $ReLU$ would have to be in $L^2(\mathbb{R})$, which it is not. On the other hand, given $x \in \mathbb{R}$ we have that

$$\sum_{k \in \mathbb{Z}} ReLU(x+k) = \sum_{k \geq -x, k \in \mathbb{Z}} ReLU(x+k) = \sum_{k \geq -x, k \in \mathbb{Z}} (x+k) = \infty$$

showing that there is no way $ReLU$ could satisfy the partition of unity condition.

A similar proof shows that translates of sine cannot form a Riesz/weak Riesz basis. $\square$

### A.1.1 Results on the error kernel and PUC condition

We recall from sec. 2.3 that the understanding of the sampling properties of the shifted basis functions $F_k$ comes down to analysing the error kernel $E_{\widetilde{F}, F}$. The reason being was that the average error $\overline{\epsilon}_s(T)^2$ is a good predictor of the true error $\epsilon_s(T)^2$.

We sketch a proof showing that the vanishing of the error kernel in the limit $T \to 0$ for a suitable test function $\widetilde{F}$ is equivalent to $F$ satisfying the partition of unity condition. We will do this under two assumptions:

A1. The Fourier transform of $F$ is continuous at $0$.

A2. The Fourier transform of $\widetilde{F}$ is continuous at $0$.

A3. The sampled signal $s$ we wish to reconstruct is contained in $W_2^r$ for some $r > \frac{1}{2}$. This assumption is needed so that the quantity $\epsilon_{corr}$ goes to zero as $T \to 0$.

We remark that an explicit construction of $\widetilde{F}$ will be given after the proof as during the course of the proof we will see what conditions we need to impose for the construction of $\widetilde{F}$ from $F$.

From the definition of the approximation operator, equation 3, we have that

$$\lim_{T \to 0} ||f - A_T(f)||_{L^2}^2 = \lim_{T \to 0} \int_{-\infty}^{\infty} E_{\widetilde{F}, F}(T\omega) |\hat{s}(\omega)|^2 \frac{d\omega}{2\omega} \tag{21}$$

where we remind the reader that the error kernel $E_{\widetilde{F}, F}$ is given by equation equation 6. We now observe that if $\widetilde{F}$ is a function such that $\hat{\widetilde{F}}$ is bounded and $F$ satisfies the first Riesz condition, condition 1 from defn. 2.1, then by definition it follows that $E_{\widetilde{F}, F}$ is bounded. Therefore in the above integral we can apply the dominated convergence theorem and compute

$$\lim_{T \to 0} ||s - A_T(s)||_{L^2}^2 = \int_{-\infty}^{\infty} \lim_{T \to 0} E_{\widetilde{F}, F}(T\omega) |\hat{s}(\omega)|^2 \frac{d\omega}{2\omega} \tag{22}$$

$$= E_{\widetilde{F}, F}(0) \int_{-\infty}^{\infty} |\hat{s}(\omega)|^2 \frac{d\omega}{2\omega} \tag{23}$$

$$= E_{\widetilde{F}, F}(0) ||s||^2 \tag{24}$$

where to get the second equality we have used assumptions A1 and A2 above and to get the third equality we have used the fact that the Fourier transform is an isometry from $L^2(\mathbb{R})$ to itself.

We thus see that the statement $\lim_{T \to 0} ||s - Q_T(s)||_{L^2}^2 = 0$ is equivalent to $E_{\widetilde{F}, F}(0) = 0$. From equation 6 this is equivalent to

$$E_{\widetilde{F}, F}(0) = |1 - \hat{\widetilde{F}}(0)\hat{F}(0)|^2 + |\hat{\widetilde{F}}(0)|^2 \sum_{k \neq 0} |\hat{F}(2\pi k)|^2 = 0. \tag{25}$$

We see that $E_{\widetilde{F}, F}(0)$ is a sum of positive terms and hence will vanish if and only if all the terms in the summands vanish. Looking at the first summand we see that we need $\hat{\widetilde{F}}(0)\hat{F}(0) = 1$, which can hold if and only if both factors are not zero. We normalise the function $F$ so that $\hat{F}(0) = \int F(x)dx = 1$. Thus the conditions that need to be satisfied are

$$\hat{\widetilde{F}}(0) = 1 \text{ and } \sum_{k \neq 0} |\hat{F}(2\pi k)|^2 = 0. \tag{26}$$

We can rewrite the second condition in equation 26 as

$$\hat{F}(2\pi k) = \delta_k \tag{27}$$

where $\delta$ denotes the Dirac delta distribution. From this viewpoint we then immediately have that the second condition can be written in the form

$$\sum_k F(x + k) = 1 \tag{28}$$

which is precisely the partition of unity condition.

The function $\widetilde{F}$ is easy to choose. Let $\mathcal{S}$ denote Schwartz space of Schwartz functions in $L^2(\mathbb{R})$. It is well known that this space is dense in $L^2(\mathbb{R})$ and that the Fourier transform maps $\mathcal{S}$ onto itself. Therefore, in the Fourier domain let $\widetilde{\mathcal{S}}$ denote the set of Schwartz functions $f$ such that $\hat{f}(0) \neq 0$. Note that $\widetilde{S}$ is dense in $L^2(\mathbb{R})$ and elements in $\widetilde{S}$ are continuous at the origin. In order to define $\widetilde{F}$ we simply take any element $f \in \widetilde{S}$ and let $\widetilde{F} = \frac{1}{\hat{f}(0)} f$. In fact, if we denote the space $\overline{\mathcal{S}}$ to consist of those Schwartz functions $f$ whose Fourier transform satisfies $\hat{f}(0) = 1$, then it is easy to see that $\overline{\mathcal{S}}$ is dense in $L^2(\mathbb{R})$. Thus the space $\overline{\mathcal{S}}$ can be used as a test space for $\widetilde{F}$ and is the defining test space for the approximation operator $A_T$.

## A.2 WHAT DOES THE PARTITION OF UNITY CONDITION MEAN?

In the previous sec. A.1.1 we saw that the vanishing of the error kernel $E_{\widetilde{F}, F}$ in the limit $T \to 0$ was equivalent to the function $F \in L^2(\mathbb{R})$ satisfying the partition of unity condition. In this section we want to explain in a more qualitative manner what the partition of unity condition means for reconstruction in the space $L^2(\mathbb{R})$.

Fix a function $F \in L^2(\mathbb{R})$, we have seen we can create the subspace $V(F) \subseteq L^2(\mathbb{R})$. For the time being let us only assume $F$ satisfies the first condition of being a Riesz basis. Recall this means that:

$$A||a||_{l^2}^2 \leq \left|\left| \sum_{k \in \mathbb{Z}} a(k) F_k \right|\right|^2 \leq B||a||_{l^2}^2, \forall a(k) \in l^2(\mathbb{R}) \tag{29}$$

Given an arbitrary function $g \in V(F)$ the above condition 29 means that when we express

$$g = \sum_{k=-\infty}^{\infty} a(k) F(x - k), \tag{30}$$

the coefficients $a(k)$ are uniquely determined. This follows because condition 29 implies that the translates $F(x - k)$ form a linearly independent set inside $V(F)$. Thus condition 1 is there to tell us how to approximate functions within $V(F)$. It states that we can perfectly reconstruct any function in $V(F)$ using the translates $\{F(x - k)\}$.

However, let us now assume that we are given a function $g \in L^2(\mathbb{R}) - V(F)$, that is $g$ is a square integrable function that does not reside in the space $V(F)$. A natural question that arises is **can we we still use elements in the space $V(F)$ to approximate $g$?** Mathematically, what this question is asking is if we are given a very small $\epsilon > 0$ can we find a function $G \in V(F)$ such that

$$||G - g||_{L^2} < \epsilon? \tag{31}$$

This is precisely where the partition of unity condition comes in:

$$\sum_{k \in Z} F(x + k) = 1, \forall x \in \mathbb{R} (\textbf{PUC}) \tag{32}$$

Mathematically, the reason the partition of unity condition is able to bridge the gap between $V(F)$ and $L^2(\mathbb{R})$ is that if we have an arbitrary function $g \in L^2(\mathbb{R}) - V(F)$, then we can write

$$g = g - G + G \tag{33}$$

for any function $G \in V(F)$. The question now is does there exist a $G \in V(F)$ that makes the quantity $g - G$ very small in the $L^2$-norm? In other words, given a very small $\epsilon > 0$ can we make $g - G$ smaller than $\epsilon$ in the $L^2$-norm.

The way to answer this question is to first note that there is a simple way to try to construct such a $G$. Namely, project $g$ onto the subspace $V(F)$ forming the function $\mathcal{P}(g) \in V(F)$. Then look at the difference

$$g - \mathcal{P}(g) \tag{34}$$

and ask can it be made very small? In general this technique does not work. However, there is another projection. Namely, we can project $g$ onto the $\Omega$-scaled signal space $V_\Omega(F)$ for $\Omega > 0$ forming $\mathcal{P}_\Omega(g)$ and ask if the difference $g - \mathcal{P}_\Omega(g)$ can be made very small. For the definition of the $\Omega$-scaled signal space $V_\Omega(F)$ please see sec. 2.3.

The partition of unity condition says that there exists a $\Omega > 0$ such that the difference

$$g - \mathcal{P}_\Omega(g) \tag{35}$$

can be made very small.

Thus the second condition from the Riesz basis definition, the partition of unity condition, is telling us how to approximate functions outside of $V(F)$ using the translates $\{F(x - k)\}$ and the scaled signal spaces $V_\Omega$. It says that we cannot necessarily perfectly reconstruct a function outside of $V(F)$ but we can reconstruct it up to a very small error using the $\Omega$-scaled signal space $V_\Omega(F)$. The partition of unity condition bridges the gap between $V(F)$ and $L^2(\mathbb{R})$ via the scaled signal spaces $V_\Omega(F)$ telling us that reconstruction is possible only in $V_\Omega(F)$ for some $\Omega > 0$.

For a full mathematical proof of how the partition of unity does this we kindly ask the reader to consult sec. A.1.1.

Let us summarize what we have discussed:

1. The first condition of a Riesz basis is there so that we know that translates of $F$ namely $\{F(x - k)\}$ can be used to uniquely approximate functions in the signal space $V(F)$. In this case, theoretically the translates $\{F(x - k)\}$ provide a perfect reconstruction.

2. The second condition of a Riesz basis, namely the partition of unity condition, is there so that we know how to approximate functions that do not lie in $V(F)$. It says that in order to bridge the gap between $V(F)$ and $L^2(\mathbb{R})$ we need to do so by going through an $\Omega$-scaled signal space $V_\Omega(F)$ for a $\Omega > 0$. In the scaled signal space perfect reconstruction is not possible but we can reconstruct up to a very small error.

### A.2.1 PROOFS OF MAIN RESULTS IN SECTION 2.3

***Proof of theorem 2.4.*** We first note that by condition 1 in defn. 2.1. The space $V(F)$ is a subspace of $L^2(\mathbb{R})$. Therefore, the space $V(F)$ with the induced $L^2$-norm forms a well-defined normed vector space.

Since $g \in V(F)$ we can write $g = \sum_{k=-\infty}^{\infty} a(k)F(x-k)$ in $L^2(\mathbb{R})$. This means that the difference

$$g - \sum_{k=-\infty}^{\infty} a(k)F(x-k) = 0 \in L^2(\mathbb{R}) \tag{36}$$

and in particular that the partial sums

$$S_n := \sum_{k=-n}^{n} a(k)F(x-k) \tag{37}$$

converge in $L^2$ to $g$ as $n \to \infty$. Writing this out, this means that given any $\epsilon > 0$, there exists an integer $k(\epsilon)$ such that

$$\left\| g - \sum_{k=-k(\epsilon)}^{k(\epsilon)} a(k)F(x-k) \right\|_{L^2} < \epsilon. \tag{38}$$

We can then define a 2-layer neural network $f$ with $n(\epsilon) = 2k(\epsilon)$ neurons as follows: Let the weights in the first layer be the constant vector $[1, \cdots, 1]^T$ and the associated bias to be the vector $[-k(\epsilon), -k(\epsilon) + 1, \ldots, k(\epsilon)]^T$. Let the weights associated to the second layer be the vector $[a(-k(\epsilon)), a(-k(\epsilon) + 1), \cdots, a(k(\epsilon))]$ and the associated bias be 0. These weights and biases will make up the parameters for the neural network $f$ and in the hidden layer we take $F$ as the non-linearity.

Applying equation 38 we obtain that

$$\|f(\theta) - g\|_L^2 < \epsilon. \tag{39}$$

$\square$

***Proof of prop. 2.9.*** The proof of this proposition will be in two steps. The reason for this is that we need to use thm. 2.7 and in doing so we want to know that the error $\epsilon_{corr}$ can be made arbitrarily small. Thm. 2.7 shows that if we assume our signal $s \in W_2^1(\mathbb{R})$, then we have the bound

$$\epsilon_{corr} \le \gamma \Omega \|s^{(1)}\|_{L^2} \tag{40}$$

where $s^{(1)}$ denotes the first Sobolev derivative, which exists because of the assumption that $s \in W_2^1$.

We thus see that if we choose $\Omega > 0$ sufficiently small we can make $\epsilon_{corr} < \frac{\epsilon}{2}$, by equation 40. Furthermore, by lemma 2.8 we have that the average approximation error $\overline{\epsilon}(\Omega) < \frac{\epsilon}{2}$ for $\Omega$ sufficiently small. Therefore, by taking $f_\Omega = A_\Omega(s) \in V_\Omega(F)$ the proposition follows for the signal $s \in W_2^1$.

As we have only proved the proposition for signals in $s \in W_2^1(\mathbb{R})$ we are not done. We want to prove it for signals $s \in L^2(\mathbb{R})$. This is the second step, which proceeds as follows.

We start by observing that $A_\Omega$ is a bounded operator from $L^2(\mathbb{R})$ into $L^2(\mathbb{R})$, see Blu & Unser (1999). Let $T = \|A_\Omega\|_{op}$ denote the operator norm of $A_\Omega$. We also use the fact that $C_c^\infty(\mathbb{R})$ is dense in $L^2(\mathbb{R})$, see Stein & Shakarchi (2011).

Then by density of $C_c^\infty(\mathbb{R})$ in $L^2(\mathbb{R})$ we can find an $f \in C_c^\infty(\mathbb{R})$ such that

$$\|f - s\|_{L^2} < \min \left\{ \frac{\epsilon}{3T}, \frac{\epsilon}{3} \right\} \tag{41}$$

$\|f - s\|_{L^2} < \frac{\eta}{3T}$. Furthermore, since $f \in C_c^\infty$ it lies in $W_2^1$. By the above we have that there exists $\Omega > 0$ such that $\|f - A_\Omega(f)\|_{L^2} < \frac{\epsilon}{3}$. We then estimate:

$$\|s - A_\Omega(s)\| = \|s - f + f - A_\Omega(f) + A_\Omega(f) - A_\Omega(s)\|_{L^2} \tag{42}$$

$$\le \|s - f\|_{L^2} + \|f - A_\Omega(f)\|_{L^2} + \|A_\Omega(f) - A_\Omega(s)\|_{L^2} \tag{43}$$

$$\le \|s - f\|_{L^2} + \|f - A_\Omega(f)\|_{L^2} + \|A_\Omega\|_{op} \|s - f\|_{L^2} \tag{44}$$

$$\le \frac{\epsilon}{3} + \frac{\epsilon}{3} + \frac{\epsilon}{3} \tag{45}$$

$$= \epsilon \tag{46}$$

where equation 43 follows from the triangle inequality and equation 44 from equation 41. This completes the proof. $\square$

***Proof of thm. 2.10.*** By prop. 2.9 there exists an $\Omega > 0$ sufficiently small and an $f_\Omega \in V_\Omega(F)$ such that

$$||s - f_\Omega||_{L^2} < \frac{\epsilon}{2}. \tag{47}$$

As $f_\Omega$ lies in $V_\Omega(F)$ we can write $f_\Omega = \sum_{k=-\infty}^{\infty} a_\Omega(k) F(\frac{1}{\Omega}(x - \Omega k)$. This implies that the partial sums

$$S_n = \sum_{k=-n}^{n} a_\Omega(k) F(\frac{1}{\Omega}(x - \Omega k) \tag{48}$$

converge under the $L^2$-norm to $f_\Omega$ as $n \to \infty$. By definition of convergence this means given any $\epsilon > 0$ there exists an integer $k(\epsilon) > 0$ such that

$$\left|\left| f_\Omega - \sum_{k=-k(\epsilon)}^{k(\epsilon)} a_\Omega(k) F\left(\frac{1}{\Omega}(x - \Omega k)\right) \right|\right| < \frac{\epsilon}{2}. \tag{49}$$

We define a neural network $\mathcal{N}$ with $n(\epsilon) = 2k(\epsilon)$ neurons in its hidden layer as follows. The weights in the first layer will be the constant vector $[1, \ldots, 1]^T$ and the associated bias will be the vector $[-\Omega k(\epsilon), -\Omega k(\epsilon) + 1, \ldots, \Omega k(\epsilon)]^T$. The weights associated to the second layer will be $[a(-k(\epsilon)), \ldots, a(k(\epsilon))]$ and the bias for this layer will be 0. These weights and biases will make up the parameters $\theta$ for the neural network. In the hidden layer we take as activation the function $F_\Omega$. With these parameters and activation function, we see that

$$\mathcal{N}(\theta)(x) = \sum_{k=-k(\epsilon)}^{k(\epsilon)} a_\Omega(k) F\left(\frac{1}{\Omega}(x - \Omega k)\right). \tag{50}$$

We then have that equation 49 implies that

$$||\mathcal{N}(\theta) - f_\Omega||_{L^2} < \frac{\epsilon}{2}. \tag{51}$$

Combining this with equation 47 we have

$$||\mathcal{N}(\theta) - s||_{L^2} = ||\mathcal{N}(\theta) - f_\Omega + f_\Omega - s||_{L^2} \tag{52}$$
$$\leq ||\mathcal{N}(\theta) - f_\Omega||_{L^2} + ||f_\Omega - s||_{L^2} \tag{53}$$
$$\leq \frac{\epsilon}{2} + \frac{\epsilon}{2} \tag{54}$$
$$= \epsilon \tag{55}$$

where equation 53 follows from the triangle inequality and equation 54 follows by using equation 47 and equation 51. The theorem has been proved.

$\square$

## B  ON TAKEN'S EMBEDDING THEOREM

Taken's embedding theorem is a delay embedding theorem giving conditions under which the strange attractor of a dynamical system can be reconstructed from a sequence of observations of the phase space of that dynamical system.

The theorem constructs an embedding vector for each point in time

$$x(t_i) = [x(t_i), x_(t_i + n\Delta t), \ldots, x(t_i + (d-1)n\Delta t)]$$

Where $d$ is the embedding dimension and $n$ is a fixed value. The theorem then states that in order to reconstruct the dynamics in phase space for any $n$ the following condition must be met

$$d \geq 2D + l$$

where $D$ is the box counting dimension of the strange attractor of the dynamical system which can be thought of as the theoretical dimension of phase space for which the trajectories of the system do not overlap.

**Drawbacks of the theorem:** The theorem does not provide conditions as to what the best $n$ is and in practise when $D$ is not known it does not provide conditions for the embedding dimension $d$. The quantity $n\Delta t$ is the amount of time delay that is being applied. Extremely short time delays cause the values in the embedding vector to almost be the same, and extremely large time delays cause the value to be uncorrelated random variables. The following papers show how one can find the time delay in practise Kim et al. (1999); Small (2005). Furthermore, in practise estimating the embedding dimension is often done by a false nearest neighbours algorithm Kennel et al. (1992).

Thus in practise time delay embeddings for the reconstruction of dynamics can require the need to carry further experiments to find the best time delay length and embedding dimension.

## C    RELATION TO UNIVERSAL APPROXIMATION

Thms. 2.4 and 2.10 can be interpreted as universal approximation theorems for signals in $L^2(\mathbb{R})$. The classic universal approximation theorems are generally for functions on bounded domains . In 92' W. A Light extended those results on bounded domains to a universal approximation for continuous function on $\mathbb{R}^n$ by sigmoid activated networks Light (1992). His result can also be made to hold for sinc activated networks, and since the space of continuous functions is dense in $L^2(\mathbb{R})$ his proof easily extends to give a universal approximation result for sinc activated 2 layer networks for signals in $L^2(\mathbb{R})$. Thus thm. 2.10 can be seen as giving a different proof of W.A. Light's result.

Although it seems like such results have been known through classical methods, we would like to emphasize that the importance of thm. 2.10 comes in how it relates to sampling theory. Given a signal $s \in L^2(\mathbb{R})$ that is bandlimited, the Nyquist-Shannon sampling theorem. This classical theorem, see Marks (2012), allows signal reconstruction using shifted sinc functions while explicitly specifying the coefficients of these shifted sinc functions. These coefficients correspond to samples of the signal, represented as $s(n/2\Omega)$. In cases where the signal is not bandlimited, prop. 2.9 still enables signal reconstruction via shifted sinc functions, albeit without a closed formula for the coefficients involved. This is precisely where thm. 2.10 demonstrates its significance. The theorem reveals that the shifted sinc functions constituting the approximation can be encoded using a two-layer sinc-activated neural network. Notably, this implies that the coefficients can be learned as part of the neural network's weights, rendering such a sinc-activated network exceptionally suited for signal reconstruction in the $L^2(\mathbb{R})$ space. In fact, thm. 2.10 shows that one does not need to restrict to sinc functions and that any activation that forms a Riesz basis will be optimal.

## D    DYNAMICAL EQUATIONS

**Lorentz System:** For the Lorenz system we take the parameters, $\sigma = 10$, $\rho = 28$ and $\beta = \frac{8}{3}$. The equations defining the system are:

$$\frac{dx}{dt} = \sigma(-x + y) \tag{56}$$

$$\frac{dy}{dt} = -xz + \rho x - y \tag{57}$$

$$\frac{dz}{dt} = -xy - \beta z \tag{58}$$

**Van der Pol Oscillator:** For the Van der Pol oscillator we take the parameter, $\mu = 1$. The equations defining the system are:

$$\frac{dx}{dt} = \mu(x - \frac{1}{3}x^3 - y) \tag{59}$$

$$\frac{dy}{dt} = \frac{1}{\mu}x \tag{60}$$

**Chen System:** For the Chen system we take the parameters, $\alpha = 5$, $\beta = -10$ and $\delta = -0.38$. The equations defining the system are:

$$\frac{dx}{dt} = \alpha x - yz \tag{61}$$

$$\frac{dy}{dt} = \beta y + xz \tag{62}$$

$$\frac{dz}{dt} = \delta z + \frac{xy}{3} \tag{63}$$

**Rössler System:** For the Rössler system we take the parameters, $a = 0.2$, $b = 0.2$ and $c = 5.7$. The equations defining the system are:

$$\frac{dx}{dt} = -(y + z) \tag{64}$$

$$\frac{dy}{dt} = x + ay \tag{65}$$

$$\frac{dz}{dt} = b + z(x - c) \tag{66}$$

**Generalized Rank 14 Lorentz System:** For the following system we take parameters $a = \frac{1}{\sqrt{2}}$, $R = 6.75r$ and $r = 45.92$. The equations defining the system are:

$$\frac{d\psi_{11}}{dt} = -a\left(\frac{7}{3}\psi_{13}\psi_{22} + \frac{17}{6}\psi_{13}\psi_{24} + \frac{1}{3}\psi_{31}\psi_{22} + \frac{9}{2}\psi_{33}\psi_{24}\right) - \sigma\frac{3}{2}\psi_{11} + \sigma a\frac{2}{3}\theta_{11} \tag{67}$$

$$\frac{d\psi_{13}}{dt} = a\left(-\frac{9}{19}\psi_{11}\psi_{22} + \frac{33}{38}\psi_{11}\psi_{24} + \frac{2}{19}\psi_{31}\psi_{22} - \frac{125}{38}\psi_{31}\psi_{24}\right) - \sigma\frac{19}{2}\psi_{13} + \sigma a\frac{2}{19}\theta_{13} \tag{68}$$

$$\frac{d\psi_{22}}{dt} = a\left(\frac{4}{3}\psi_{11}\psi_{13} - \frac{2}{3}\psi_{11}\psi_{31} - \frac{4}{3}\psi_{13}\psi_{31}\right) - 6\sigma\psi_{22} + \frac{1}{3}\sigma a\theta_{22} \tag{69}$$

$$\frac{d\psi_{31}}{dt} = a\left(\frac{9}{11}\psi_{11}\psi_{22} + \frac{14}{11}\psi_{13}\psi_{22} + \frac{85}{22}\psi_{13}\psi_{24}\right) - \frac{11}{2}\sigma\psi_{31} + \frac{6}{11}\sigma a\theta_{31} \tag{70}$$

$$\frac{d\psi_{33}}{dt} = a\left(\frac{11}{6}\psi_{11}\psi_{24}\right) - \frac{27}{2}\sigma\psi_{33} + \frac{2}{9}\sigma a\theta_{33} \tag{71}$$

$$\frac{d\psi_{24}}{dt} = a\left(-\frac{2}{9}\psi_{11}\psi_{13} - \psi_{11}\psi_{33} + \frac{5}{9}\psi_{13}\psi_{31}\right) - 18\sigma\psi_{24} + \frac{1}{9}\sigma a\theta_{24} \tag{72}$$

$$\frac{d\theta_{11}}{dt} = a\Bigg(\psi_{11}\theta_{02} + \psi_{13}\theta_{22} - \frac{1}{2}\psi_{13}\theta_{24} - \psi_{13}\theta_{02} + 2\psi_{13}\theta_{04} + \psi_{22}\theta_{13} + \psi_{22}\theta_{31} + \psi_{31}\theta_{22} \tag{73}$$

$$+ \frac{3}{2}\psi_{33}\theta_{24} - \frac{1}{2}\psi_{24}\theta_{13} + \frac{3}{2}\psi_{24}\theta_{33}\Bigg) + Ra\psi_{11} - \frac{3}{2}\theta_{11}$$

$$\frac{d\theta_{13}}{dt} = a\Bigg(-\psi_{11}\theta_{22} + \frac{1}{2}\psi_{11}\theta_{24} - \psi_{11}\theta_{02} + 2\psi_{11}\theta_{04} - \psi_{22}\theta_{11} - 2\psi_{31}\theta_{22} \tag{74}$$

$$+ \frac{5}{2}\psi_{31}\theta_{24} + \frac{1}{2}\psi_{24}\theta_{11} + \frac{5}{2}\psi_{24}\theta_{31}\Bigg) + Ra\psi_{13} - \frac{19}{2}\theta_{13}$$

$$\frac{d\theta_{22}}{dt} = a\Bigg(\psi_{11}\theta_{13} - \psi_{11}\theta_{31} - \psi_{13}\theta_{11} + 2\psi_{13}\theta_{31} + 4\psi_{22}\theta_{04} - \psi_{33}\theta_{11} + 2\psi_{24}\theta_{02}\Bigg) + 2Ra\psi_{22} - 6\theta_{22} \tag{75}$$

$$\frac{d\theta_{31}}{dt} = a\Bigg(\psi_{11}\theta_{22} - 2\psi_{13}\theta_{22} + \frac{5}{2}\psi_{13}\theta_{24} - \psi_{22}\theta_{11} + 2\psi_{22}\theta_{13} + 4\psi_{31}\theta_{02} - 4\psi_{33}\theta_{02} \tag{76}$$

$$+ 8\psi_{33}\theta_{04} - \frac{5}{2}\psi_{24}\theta_{13}\Bigg) + 3Ra\psi_{31} - \frac{11}{2}\theta_{31}$$

$$\frac{d\theta_{33}}{dt} = a\left(\frac{3}{2}\psi_{11}\theta_{24} - 4\psi_{31}\theta_{02} + 8\psi_{31}\theta_{04} - \frac{3}{2}\psi_{24}\theta_{11}\right) + 3Ra\psi_{33} - \frac{27}{2}\theta_{33} \tag{77}$$

$$\frac{d\theta_{24}}{dt} = a\Bigg(\frac{1}{2}\psi_{11}\theta_{13} - \frac{3}{2}\psi_{11}\theta_{33} + \frac{1}{2}\psi_{13}\theta_{11} - \frac{5}{2}\psi_{13}\theta_{31} - 2\psi_{22}\theta_{02} \tag{78}$$

$$- \frac{5}{2}\psi_{31}\theta_{13} - \frac{3}{2}\psi_{33}\theta_{11}\Bigg) + 2Ra\psi_{24} - 18\theta_{24}$$

$$\frac{d\theta_{02}}{dt} = a\Bigg(-\frac{1}{2}\psi_{11}\theta_{11} + \frac{1}{2}\psi_{11}\theta_{11} + \frac{1}{2}\psi_{11}\theta_{13} + \frac{1}{2}\psi_{13}\theta_{11} + \psi_{22}\theta_{24} \tag{79}$$

$$- \frac{3}{2}\psi_{31}\theta_{31} + \frac{3}{2}\psi_{31}\theta_{33} + \frac{3}{2}\psi_{33}\theta_{31} + \psi_{24}\theta_{24}\Bigg) - 4\theta_{02}$$

$$\frac{d\theta_{04}}{dt} = -a\Bigg(\psi_{11}\theta_{13} + \psi_{13}\theta_{11} + 2\psi_{22}\theta_{22} + 4\psi_{31}\theta_{33} + 4\psi_{33}\theta_{31}\Bigg) - 16\theta_{04} \tag{80}$$

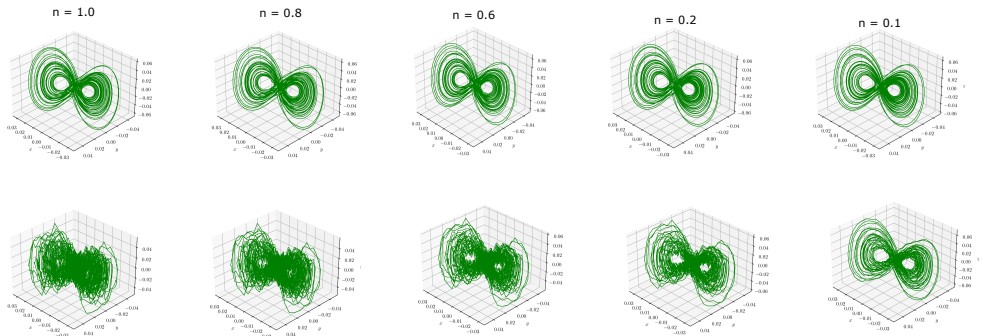

Figure 4: Robust recovery of dynamical systems from partial observations (Lorenz system). *Top row:* coordinate network. *Bottom row:* classical method.

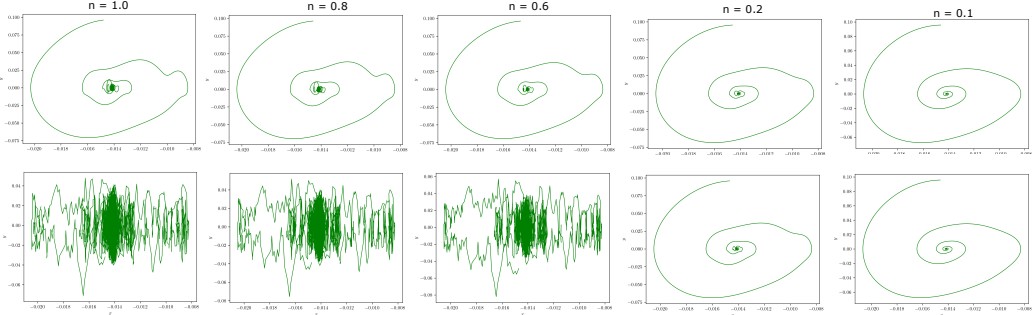

Figure 5: Robust recovery of dynamical systems from partial observations (Duffing system). *Top row:* coordinate network. *Bottom row:* classical method.

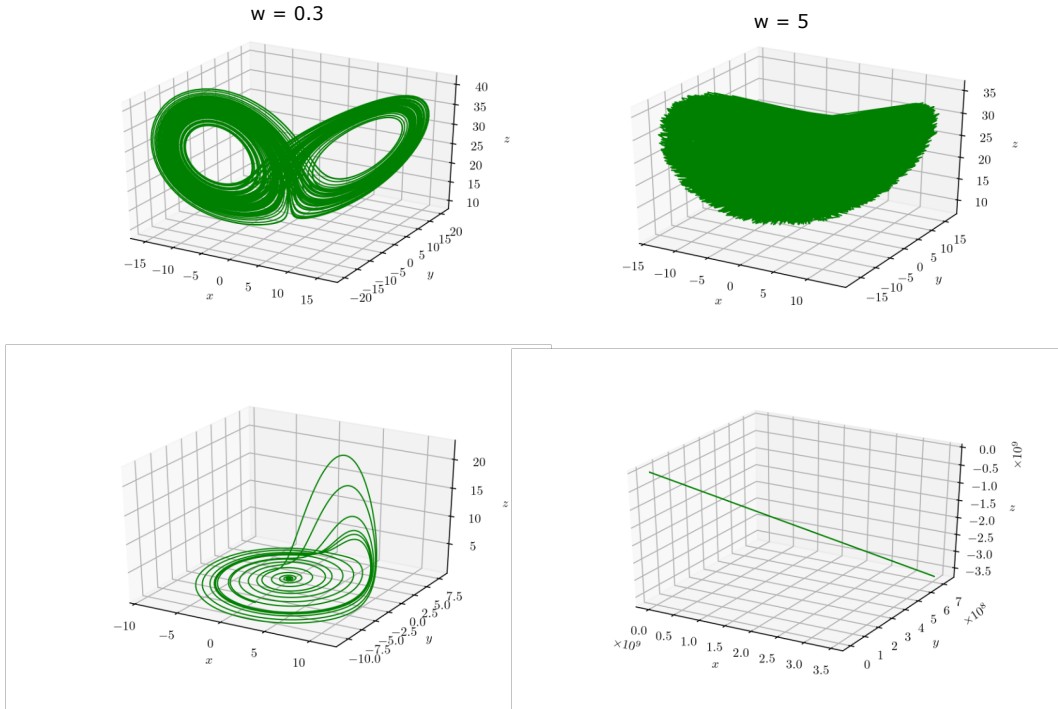

Figure 6: The top row and the bottom row depicts the SINDy reconstructions obtained for the Lorenz system and the Rossler system, respectively, using coordinate networks. As $\omega$ is increased in the sinc function, the coordinate network allows more higher frequencies to be captured, resulting in noisy reconstructions.

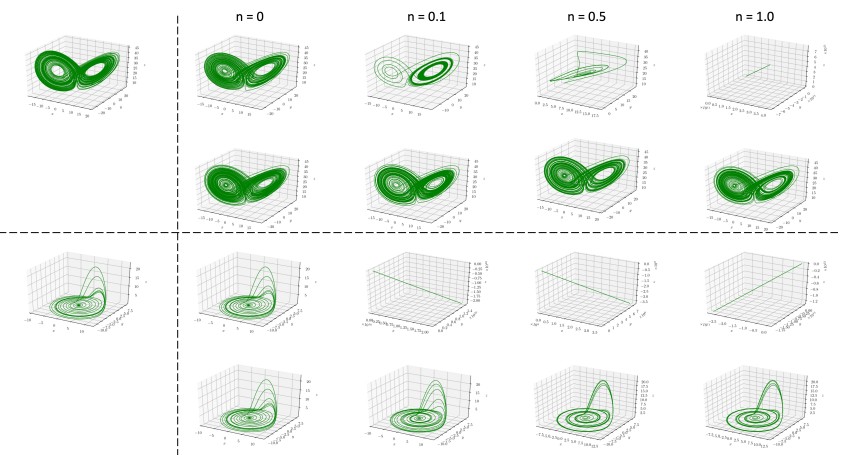

Figure 7: We use sinc-INRs to improve the results of the SINDy algorithm. The top block and the bottom block demonstrate experiments on the Lorenz system and the Rossler system, respectively. In each block, the top row and the bottom row represent the results of the baseline SINDy algorithm and the improved version (using coordinate networks). As evident, coordinate networks can be used to obtain significantly robust results.

# E    SINC ACTIVATIONS FOR POSITIONAL EMBEDDINGS

Recent research, notably Zheng et al. (2022), has provided compelling evidence that the effectiveness of positional encodings need not be exclusively tied to a Fourier perspective. They demonstrate that non-Fourier embedding functions, such as shifted Gaussian functions, can be effectively utilized for positional encoding. These functions are characterized by having a sufficiently high Lipschitz constant and the ability to generate high-rank embedding matrices, attributes that are shown to achieve results comparable to Random Fourier Feature (RFF) encodings.

Building on this, research in Ramasinghe & Lucey (2023) further confirms that shifted Gaussian functions with spatially varying variances can surpass the performance of RFF encodings. Given that sinc functions also exhibit these desirable properties, they can be feasibly employed as shifted basis functions for high-frequency signal encoding.

To explore this, we developed a sinc-based positional embedding layer. For a 2D coordinate $(x_1, x_2)$, each dimension is embedded using sinc functions:

$$\psi_1(x_1) = [\text{sinc}(\|t_1 - x_1\|), \text{sinc}(\|t_2 - x_1\|), \ldots, \text{sinc}(\|t_N - x_1\|),$$

$$\psi_1(x_2) = [\text{sinc}(\|t_1 - x_2\|), \text{sinc}(\|t_2 - x_2\|), \ldots, \text{sinc}(\|t_N - x_2\|),$$

where $t_1, \ldots, t_N$ are equidistant samples in $[0, 1]$. Then, these embeddings are concatenated to create the final embedding as,

$$\Psi(x_1, x_2) = [\psi_1(x_1), \psi_1(x_2)]$$

In a comparative study using the DIV2K dataset for image reconstruction, our sinc-based positional embedding layer demonstrated superior performance to an RFF-based layer, as shown Table 3:

| PE layer | PSNR |
|----------|------|
| RFF      | 23.5 |
| Sinc PE  | 26.4 |

Table 3: Comparison of the sinc psitional embedding layer against RFF positional embeddings.

This result indicates that sinc-based positional embeddings offer a promising alternative to RFF encodings.

