# A   SAMPLING THEORY IN HIGHER DIMENSIONS

The sampling theory — in its original form — is only applicable to one dimensional signals. However, it can be extended to higher dimensions in a straightforward manner. Let $f : \mathbb{R}^n \to \mathbb{R}$ be a function in $L^1(\mathbb{R}^n)$, which we think of as a higher mode signal. Let $I(\Omega_1, \ldots, \Omega_n)$ denote an n-dimensional rectangle about the origin with side lengths $\Omega_1, \ldots, \Omega_n$. Suppose that the Fourier transform $\widehat{f}$ vanishes identically outside of $I(\Omega_1, \ldots, \Omega_n)$. Then

$$f(t_1, \ldots, t_n) = \sum_{m_1=-\infty}^{\infty} \cdots \sum_{m_n-\infty}^{\infty} f\left(\frac{m_1}{2\Omega_1}, \ldots, \frac{m_n}{2\Omega_n}\right) sinc\left(2\Omega_1\left(t_1 - \frac{n}{2\Omega_1}\right)\right) \cdots sinc\left(2\Omega_n\left(t_n - \frac{n}{2\Omega_n}\right)\right).$$

Thus we see that sampling $f$ on the lattice defined by lengths $\left(\frac{1}{2\Omega_1}, \ldots, \frac{1}{2\Omega_n}\right)$ and taking shifted sinc functions of bandwidth $2\Omega_k$, for $1 \le k \le n$, we can reconstruct the function $f$ as in the one dimensional case. Note that as in the case of the one-dimensional Nyquist-Shanon theorem, in order for perfect reconstruction one needs to sample at larger than twice the dominant frequency present in the signal. Therefore, in practise one would take the maximum of $\Omega = \max_i\{\Omega_i\}$ and sample at a frequency of $2\Omega$.

**Curse of dimensionality.** While the multidimensional Nyquist-Shanon sampling theorem provides a convenient theoretical framework in which to understand signal processing problems in higher dimensions. It does not come without problems. In practise, the multidimensional sampling theorem is extremely inefficient.

The main issue with sampling in higher dimensions is that there is an exponential increase in volumes of cubes (or rectangles/balls) associated with adding extra dimensions. To see this, imagine we had a signal $f : [0, 1] \to \mathbb{R}$ whose dominant frequency was 50-Hertz. Let us then suppose we wish to perform a reconstruction by using a sample rate of 100-Hertz. This means that we would need to sample exactly $10^2 = 100$ points from the unit interval $[0, 1]$ each spaced at a distance of 0.01. Now, imagine that we had a 10 mode signal $g : [0, 1]^{10} \to \mathbb{R}$ on the unit cube whose dominant frequency was also 50-Hertz. We wish to perform a 100-Hertz sample rate reconstruction of $g$ as we did for $f$. Now we see a problem, in this instance we would need to sample $(10^2)^10 = 10^20$ points from the 10-dimensional cube. Thus when using a sampling distance of 0.01 we see that the 10-dimensional cube $[0, 1]^10$ is $10^{18}$-times larger than the 1-dimensional cube $[0, 1]$. This exponential increase in the amount of sample points needed to reconstruct a high mode signal is referred to as the curse of dimensionality and is a mathematical consequence of the fact that volumes of many mathematical shapes grow exponentially with dimension. This makes the sampling theory of Nqyquist and Shanon some what unusable in practise for higher mode signals.

There have been other reconstruction techniques, most notable compressed sensing, that have shown far superior performance than classical sampling due to their ability to break the Nyquist limit and allow far fewer sampling points. However, such techniques have the added problem that they are memory intensive for high mode signals. As we show INRs offer a convenient middle ground that makes them perfectly suitable for signal reconstruction in higher mode signal settings.

# B   PROOFS OF RESULTS IN SECTION 3.2

## B.1   PRELIMINARIES

We will be using the basic theory of Hilbert spaces in $L^2$. Namely, the space of square integrable functions on $\mathbb{R}$ will be denoted by $L^2(\mathbb{R})$ and we recall that this is defined as the vector space of equivalence classes of measurable functions on $\mathbb{R}$ with the following inner product

$$\langle f, g \rangle_{L^2} = \int_{\mathbb{R}} f \cdot g. \tag{14}$$

We wil also need to make use of the Sobolev spaces of order $r$, denoted by $W_2^r(\mathbb{R})$. We define this space as the space of $L^2$-functions that have $r$ weak derivatives that are also in $L^2(\mathbb{R})$.

***Proof of prop. 3.3.*** The sinc function is in $L^2$ and further since its Fourier transform is the rectangular function it is easy to see using the fact that the Fourier transform is an isometry of $L^2$ that it must satisfy the Partition of unity conditon.

The Gaussian $e^{x^2/2s^2}$ is also in $L^2$, however this function does not satisfy the partition of unity condition. The reason being that by the Poisson summatic formula it suffice to show that the translates over the integers of the Fourier transform sums to 1. However, the Fourier transform of a Gaussian is a Gaussian, therefore outside the rectangle $[-\pi/\beta, \pi/\beta]$ the exponential decay cannot contribute anything to the sum in the partition of unity condition. Hence the sum could not sum to 1. It follows the translates of the Gaussian can only form

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

. 3.9_**. We start by proving the proposition for the case that $s \in W_2^1$. We then note that in this case by thm. 3.7, there is a $\Omega > 0$ sufficiently small such that $\epsilon_{corr} < \frac{\epsilon}{2}$. Furthermore, by lemma 3.8 we have that average approximation error $\bar{\epsilon}(\Omega) < \frac{\epsilon}{2}$ for $\Omega$ sufficiently small. Therefore, by taking $f_\Omega = A_\Omega(s) \in V_\Omega(F)$ the proposition follows for signal in $W_2^1$.