# OpenReview forum: "On the Optimality of Activations in Implicit Neural Representations"
_ICLR.cc/2024/Conference — Submitted to ICLR 2024_

### Official Review · Reviewer_8ETH · 2023-10-14

**Soundness:** 3 good
**Presentation:** 2 fair
**Contribution:** 4 excellent
**Rating:** 6
**Confidence:** 3

**Summary:**

The paper views INR fitting through the lens of sampling theory and shows that the sinc activation function is particularly amenable due to two properties: shifted sinc functions form a linearly independent basis in the space of bandlimited functions, and they also form a partition of unity in this space. The authors show that these two conditions are uniquely satisfied by sinc, and that existing activation functions only satisfy at most the first property. These two conditions allow one to construct an INR with just a single hidden layer and sinc activation that can approximate any L^2 function to arbitrary accuracy. The authors empirically demonstrate that this simple method achieves superior performance on radiance field fitting and dynamical system fitting.

**Strengths:**

This paper introduces a simple and effective method, yet the theoretical grounding is highly original and strongly motivated. The experiments are also fairly strong, with an interesting and unique combination of demonstrating applications to NeRF as well as solving dynamical systems.

**Weaknesses:**

My key concern is that the paper is hard to read, especially for many potential users of this work (e.g. NeRF practitioners). When the paper involves this much theory, it is particularly important for the presentation to be clear and to offer graphics / intuition when possible. In particular, everything from def 3.5 to lemma 3.8 seems to have the sole purpose of demonstrating the importance of the PUC condition in bridging the gap between V(F) and L^2(R), but after reading it I still don't have an intuitive understanding of how the PUC condition enables this (I believe that only if I carefully followed the proofs in the appendix would I understand this). This lack of intuition also makes it difficult to think critically about the connection between the theory and behaviors of the different activations in practice. (e.g. to what extent is the performance of ReLU activations actually limited by them not forming a Riesz basis?) Since this subsection (def 3.5 to lemma 3.8) constitutes 1.5 pages of very dense material, I think the authors should move some of this to the appendix and focus on building intuition. One way of doing that would be to answer: "what happens if you try to fit a function in L^2(R) but not in V(F) with a weak Riesz basis vs. a strong basis?" and offer a simple graphical demonstration with such a toy function.
There are also a few notational problems (see questions section below for some examples) that make the theory part a little harder to read.

**Questions:**

- under def 3.1 I don't understand the step "the upper inequality in (1) implies that the L^2 norm of a signal sinV(F) is finite". how did you bring sin into this?
- almost all modern radiance field parameterizations with INRs use some form of explicit spatial representation (Instant NGP, ZipNeRF, TensoRF, DVGO, factor fields, and many more) usually with just a single hidden layer in the MLP decoder. since theorem 3.10 only requires a single hidden layer, it would be interesting to examine whether sinc activations yield improvements in those kinds of models as well

Minor typos / notational problems:
- eqn 1 should have l \in [1,...,L-1]
- right below that, "weights biases" -> "weights and biases"
- I think remembering what all the symbols mean would be much easier if you chose a convention and stuck with it, e.g. all functions are uppercase, neural networks are calligraphic letters, etc. Right now you use f at different points to denote a neural network, a function in V(F), and a Schwartz function.
- in eqn 5, it is not clear what argument the central dot is supposed to represent or where it comes from
- F with bar + hat + tilde is excessive. and what is bar?
- be consistent in how you write summation over integers (k \in Z vs. from -inf to inf)
- Fig 2 and 5 captions use excessive negative vspace
- Conclusion: "samplingh theory"

---

> ### Author Response · Authors · 2023-11-15
> **Response to Reviewer 8ETH**
>
> **My key concern is that the paper is hard to read, especially for many potential users of this work (e.g. NeRF practitioners). When the paper involves this much theory, it is particularly important for the presentation to be clear and to offer graphics / intuition when possible....**
>
> We thank the reviewer for their concern and we understand their sentiment that the paper is dense. One of the main reason we included some of theory in its abstract form in the main paper is that a key point of this paper was to develop a theoretical framework that can be used by theoreticians and practitioners in the field. In present day machine learning there is a deep divide between practise and theory and part of our goal in this paper was to bridge this gap. Therefore, we thought it reasonable to include some of the theory in main paper and back it up with experiments towards the end, showing that our theory can be used to better understand empirical observations about INRs. We have added some more explanations, references and details to various aspects of the proofs of the main theorems in the appendix in the revised version, thereby making these parts of the paper much more readable.
>
> **what happens if you try to fit a function in L^2(R) but not in V(F) with a weak Riesz basis vs. a strong basis?**
>
> We carry out this procedure in the experiments section. We consider general $L^2$ signals and compare fitting them with neural networks admitting activations such as sinc, Gaussian, sine, wavelets and ReLU. In all cases sinc does better showing that the condition that it generates a Riesz basis leads to its ability to fit a general $L^2$ signal much better than the others.
>
>
> **under def 3.1 I don't understand the step "the upper inequality in .....**
>
> Sorry, this was a typo. It should read $s \in V(F)$. Thank you for pointing this out.
>
> **Almost all modern radiance field parameterizations with INRs use some form of explicit spatial representation (Instant NGP, ZipNeRF, TensoRF, DVGO, factor fields, and many more) usually with just a single hidden layer in the MLP decoder. since theorem 3.10 only requires a single hidden layer, it would be interesting to examine whether sinc activations yield improvements in those kinds of models as well**
>
> This is an interesting suggestion. Most of the above mentioned methods use shallow networks with RFF encodings over a spatial grid. Since sinc activations can be used as a better drop in replacement for RFF-INRs, it should be possible to get better or on-par results with sinc-INRs. However, we would also like to note that in these settings, the INRs only encode a small volume in a partitioned space, resulting in low resolution signals. Therefore, the advantage we gain from using a more capable sinc-INR might be not significantly evident.
>
>
> **Minor typos / notational problems...**
>
> Thank you! All of these have been fixed.
>
> **In eqn 5, it is not clear what argument the central dot is supposed to represent or where it comes from**
>
> The central dot is just a placement for the variable of the function $s$ i.e. it can be read as $s(x - \tau)$ where $x$ is the function variable.
>
> **F with bar + hat + tilde is excessive. and what is bar?**
>
> We thank the reviewer for bringing this up. The bar notation is a typo and should not be there in equation (6), we have corrected this.
> In general, notation in functional analysis and harmonic analysis has conventions that have gone through for decades although many of them are not ideal. It is always the case that one uses hat to denote the Fourier transform and tilde to denote a distribution, test function, associated to a given function.
> As we are combining both of these, the notation can seem rather cumbersome, though it retains the key aspect that these operators, taking an associated test function and then applying a Fourier transform, are function dependent. If we replace the notation with something completely different we fear it will cause confusion amongst those who work in this space and this is why we dicided to stick with the notation.

---

> > ### Comment · Reviewer_8ETH · 2023-11-22
> > **Thanks for response**
> >
> > Thanks for the response. I am fine with the theory and experiments, my problem is that the intuition is lacking. The PUC condition bridges the gap between V(F) and L^2(R), but why is this the case? What is the partition of unity really doing in this context? Without seeing the revised version of the paper, I can't determine whether this phenomenon is sufficiently well explained after the authors' edits.

---

> ### Author Response · Authors · 2023-11-23
> **Response to reviewer 8ETH**
>
> Thank you for your response. We have added an explanation of what the partition of unity condition is doing and how it is bridging the gap between $V(F)$ and $L^2(\mathbb{R})$. The explanation is given in the appendix in section A.2. Can you please have a look at the explanation and let us know if that is helpful?
>
> Here we give a summary of the key steps:
>
> Let us assume that we are given a function
> $g \in L^2(\mathbb{R}) - V(F)$, that is $g$ is a square integrable function that does not reside in the space $V(F)$. A natural question that arises is, can we we still use elements in the space $V(F)$ to approximate $g$? Mathematically, what this question is asking is if we are given a very small $\epsilon > 0$ can we find a function $G \in V(F)$ such that
>
> $$\vert\vert G - g \vert\vert_{L^2} < \epsilon ?$$
>
> This is precisely where the partition of unity condition comes in. Mathematically, the reason the partition of unity condition is able to bridge the gap between $V(F)$ and $L^2(\mathbb{R})$ is as follows. We can write
> $$g = g - G + G$$
> for any function $G \in V(F)$. The question now is does there exist a $G \in V(F)$ that makes the quantity $g - G$ very small in the $L^2$-norm? In other words, given a very small $\epsilon > 0$ can we make $g - G$ smaller than $\epsilon$ in the $L^2$-norm.
>
> The way to answer this question is to first note that there is a simple way to try to construct such a $G$. Namely, project $g$ onto the subspace $V(F)$. Let us denote this projection by $\mathcal{P}(g)$ and note that $\mathcal{P}(g) \in V(F)$. We can then  look at the difference
> $$g - \mathcal{P}(g)$$
> and ask can it be made very small? In general this technique does not work and the projection  $\mathcal{P}(g)$ won't be close to $g$ in the $L^2$-norm.
>
> However, there is another projection. Namely, we can project $g$ onto the $\Omega$-scaled signal space $V_{\Omega}(F)$ for
> $\Omega > 0$ and form $\mathcal{P}_{\Omega}(g)$,  which lies in the scaled signal space
>
> $$V_{{\Omega}}(F)$$
>
> The partition of unity condition is precisely the condition that allows one to deduce that there exists a $\Omega > 0$ such that the difference
> $$ g - \mathcal{P}_{\Omega}(g)$$ can be made very small.
>
> Thus the second condition from the Riesz basis definition, the partition of unity condition, is telling us how to approximate functions outside of $V(F)$ using the scaled translates
> $\{F(\frac{x}{\Omega}-k)\}$ and the scaled signal spaces $V_{\Omega}(F)$. It says that we cannot necessarily perfectly reconstruct a function outside of $V(F)$ but we can reconstruct it up to a very small error using the $\Omega$-scaled signal space $V_{\Omega}(F)$. The mathematical proof of how it does this is given in sec. A.1.1 of the appendix.
>
> Another way to state what we are saying is that the partition of unity condition bridges the gap between $V(F)$ and $L^2(\mathbb{R})$ via the scaled signal spaces $V_{\Omega}(F)$ telling us that reconstruction is possible only in $V_{\Omega}(F)$ for some $\Omega > 0$.
>
> We hope that helps your understanding. Please let us know if there is anything else we should add. Furthermore, thank you so much in taking the time to read our work. It is greatly appreciated.

---

### Official Review · Reviewer_hcEo · 2023-11-01

**Soundness:** 4 excellent
**Presentation:** 3 good
**Contribution:** 3 good
**Rating:** 6
**Confidence:** 4

**Summary:**

This paper provides a detailed analysis of approximation properties of implicit neural representations (INR) by deploying tools from harmonic analysis. Specifically, while a variety of activation functions have been investigated for use within INR models, a unified treatment is missing and comparisons appear to be entirely numerical on a small set of images. This paper categorizes the current options in the literature in terms of whether or not they form a Riesz basis or weak Riesz basis. A basis based on shifted sinc functions is described, and the correspond approximability analysis involves an interesting use of older results from Unser. Applications to dynamical systems is discussed and a small set of experiments are presented.

**Strengths:**

1. Despite the existing theoretical results dealing with INRs, the paper's categorization of known results in terms of Riesz basis is interesting.
2. The approximation analysis of bandlimited signals using $\Omega$-scaled signal space, is a valuable addition to known results on INRs.
3. The theoretical analysis overall and Thm 3.4 and 3.10 in particular will likely be used by follow-up papers on the topic.
4. Several different experimental examples are considered, from image representation to dynamical systems.

**Weaknesses:**

1. Not a major weakness but the use of orthogonal polynomials (such as Hermite) has been proposed for use as activation functions, and  its convergence/approximation properties as well its empirical benefits have been demonstrated. This paper does offer more than yet another activation/basis (which is a plus) but Prop 3.3 and the text leading up to it will benefit from a clearer positioning of why the result will be incomplete or weaker unless the characterization was in terms of Riesz basis.

2.  While it is clear that the main selling point of the paper is the technical analysis, the experimental presentation can be more thoroughly presented (even if it is short). Fig. 1 would have us believe that for the dataset described, the proposed method yields the best overall performance. Very little additional discussion is provided. Nerf experiments are presented yet there is no discussion of whether the changes can be dropped into current implementations commonly used by practitioners. Dynamical systems experiments are nice but its not obvious why the vanderpol oscillator dynamics would be more challenging than those in Fig. 1.

**Questions:**

Overall I like the paper and its creative use of harmonic analysis for this problem. Please comment on the two minor weaknesses.

The paper would also benefit from a careful proof reading. See section title of 3.2, overloading of $s$ and others.

---

> ### Author Response · Authors · 2023-11-15
> **Response to reviewer hcEo**
>
> We thank the reviewer for the valuable suggestions. Please find our answers below.
>
> **Not a major weakness but the use of orthogonal polynomials (such as Hermite) has been proposed...**
>
> This is a very reasonable question from the reviewer and we thank them for asking it. It is true that practitioners in ML have found that Hermite polynomials do perform well as activations. However, this has predominately been shown on the empirical front. Our goal in this paper was to establish a theoretical framework that offers conditions that the practising machine learner should seek when designing a new activation. We found that for signal reconstruction in $L^2$ a key condition is the one of being able to generate a Riesz basis. We should say the Hermite polynomials do not generate a Riesz basis as they do not live in $L^2$ however they do form an orthonomal basis in the Gaussian measure space $L^2(\sigma)$, where $\sigma$ denotes the Gaussian measure. It would be great if the reveiwer could provide a reference for where the Hermite polynomials are being used for signal reconstruction. In the setting of implicit neural representations, they are generally not used as they underperform when compared to other activations.
>
> **While it is clear that the main selling point of the paper is the technical analysis, the experimental presentation can be more thoroughly presented (even if it is short). Fig. 1 would have us believe that for the dataset described, the proposed method yields the best overall performance. Very little additional discussion is provided. Nerf experiments are presented yet there is no discussion of whether the changes can be dropped into current implementations commonly used by practitioners.**
>
> Thank you for the suggestion to improve our paper. We have added a more clear explanation to the experiment corresponding to Figure 1. Also, we used the original settings proposed by [1] for NeRF experiments. However, we would like to point out that $\mathrm{sinc}$-INRs should be able to be used as a better drop in replacement for any INR, as we have verified across various tasks.
>
> [1] - Mildenhall, Ben, et al. "NeRF: Representing Scenes as Neural Radiance Fields for View Synthesis." European Conference on Computer Vision. Cham: Springer International Publishing, 2020.
>
> **Dynamical systems experiments are nice but its not obvious why the vanderpol oscillator dynamics would be more challenging than those in Fig. 1.**
>
> We agree that a more theoretically involved analysis can be conducted to investigate the intricate connections between the dynamics of the systems and the performance of the INRs that are used to encode them. However, we believe that precise modeling of this connection would be out of the scope of this paper.

---

> > ### Comment · Reviewer_hcEo · 2023-11-23
> > **orthogonal polynomials, Nerfs etc**
> >
> > Dear authors,
> >
> > 1. Since the paper seeks to unify alternative proposals (which exist in the literature) under a specific choice of a basis, a natural question is whether alternative classes of basis fall short. For example, function approximation with orthogonal polynomial basis is well studied (Approximation Results for Orthogonal Polynomials in Sobolev Spaces, Canuto, Quarteroni, Math. of Comp. 1982). While the current version of the paper discusses exp, sin, and wavelets etc., in terms of whether or not they classify as a Riesz basis, it is not clear from the technical analysis whether the Riesz basis is necessary. Separately, Hermite polynomials have been used to study the optimization landscape of simpler models (see Learning One-hidden-layer Neural Networks with Landscape Design, Ge, Lee, Ma) and my comment was seeking clarification on whether other common choices will not easily lead to one (or more) results described in the paper. I must clarify that my comment does not imply that what is currently included in the paper is not a meaningful result.
> >
> > 2. My opinion regarding the experiments stands (although its minor). To substantiate the drop-in replacement statement, e.g., with NERFs, benchmarks are now available that can be tested if the drop-in claims are true. There is no reason to report a table with numerical results restricted to examples presented in the original paper from 3+ years ago. If the proposed scheme doesn't work as well as alternatives, that is completely fine but a less experienced reader will have a much clearer calibration of the experimental findings, and how literally they should be taken.

---

> ### Author Response · Authors · 2023-11-23
> **Response to reviewers hcEo**
>
> Thank you very much your response and your two comments. We greatly appreciate you taking the time to give us feedback.
>
> 1. We see your point of view and agree there is merit in what you are saying. Our theory provides a sufficient condition to chose an optimal activation yet the theory does not provide necessary conditions. This is in itself an open problem and as of yet we  feel trying to show the theory is necessary will require a lot of work and we hope to take this up in a future project. However, in order to give credence to our theory, we did test it among various different types of experiments. For these experiments we purposefully chose 4 different types of functions. We chose a sine function because sine and cosine form a Fourier basis for periodic functions (note a cosine is just a shifted sine), we chose a wavelet function because wavelet functions can generate a basis for frequency space, we chose a Gaussian function because Gaussians generate a weak Riesz basis and finally we chose a sinc function because such a function generates a Riesz basis. Thus each of the functions we chose is related to a type of basis used in signal processing. We wanted to then use the experiments as a means of testing how well each of these bases do with the hindsight that a sinc function should do extremely well as our theory says that a sufficient condition for a function to perform well as an activation for INRs is if it generates a Riesz basis. Finding that the other functions did not perform as well as a sinc does not necessarily prove that the Riesz basis condition for an activation is necessary for optimal performance but it does suggest that as a sufficient condition it is extremely useful for the practising machine learner. We completely agree that this does not rule out another basis function such as Hermite functions doing equally as well yet we decided to test on sine, wavelet, Gaussian, and sinc because these are the most popular ones used within the INR community.
>
>
> 2. Thank you for the point regarding the experiments, we certainly understand what you are saying. The main reason for choosing the classical NeRF architecture was simply one of choice. Most newer NeRF architectures are enrichments of the classical NeRF architecture for a particular problem or as a means to enhance the ability of the classical NeRF. Therefore, since our approach requires focusing on the activation we felt that it be best if we went with the classical NeRF architecture from 3+ years ago as this has been used by several practitioners in the field and those using more modern architectures will easily be able to drop in our activation into their architecture as most of the newer NeRF style architectures build upon the classical one. Further, our goal was to present a more broad overview of results showing that our theory is useful not only in vision contexts but dynamical ones too. So we thought we would go with famous architectures people know about in each one.

---

### Official Review · Reviewer_REJk · 2023-11-01

**Soundness:** 3 good
**Presentation:** 2 fair
**Contribution:** 3 good
**Rating:** 5
**Confidence:** 4

**Summary:**

The authors study neural network activation functions from the perspective of Shannon-Nyquist sampling theory. This study is particularly relevant to implicit neural representations, which are usually overfitted to regularly sampled signals, e.g., the coordinate-intensity pairs of an image.

The authors prove a universal approximation theorem for 2-layer, $\mathrm{sinc}$-activated networks and note that the same theorem does not hold for ReLU, $\sin$, and Gaussian activations and could but does not necessarily hold for wavelet activations.

The authors demonstrate the practical relevance of their theory by using INRs to perform image reconstruction, construct neural radiance fields, and discover the governing dynamics of chaotic dynamical systems.

**Strengths:**

I am quite excited about the theoretical results in this paper. The authors' results offer a simple criterion for activation functions that is easy to check in practice to determine whether they will work well for INRs. I checked all proofs in detail and believe they are correct; they follow elegantly from reasonably elementary results from functional analysis. Finally, I found the author's construction for the 2-layer INRs achieving an arbitrarily small approximation gap elegantly simple.

**Weaknesses:**

Unfortunately, the brilliance of the ideas is significantly diminished by the unacceptably low quality of writing. Concretely, the paper looks and reads like a draft that is 80-90% complete.

In particular:
- There are countless typos in the paper, including undefined notation, which makes parsing the contents quite challenging. For example, I believe that in the paragraph below Eq 5, T is undefined, and I think it should be 1/Omega instead. Similarly, there's an inequality sign missing in the bound on $|\varepsilon_{corr}|$ in Thm 3.7.
- This is an even more serious problem in the appendix, where it appears that the content wasn't even re-read once, as there are gross typos (including some mathematical ones) in every second to third sentence. For example:
  - Appendix A (which is also unreferenced in the main text) has several typos of the form $10^20$ instead of $10^{20}$
  - The title of Appendix B is "proof of theorems in Sec 3.2", but the theorems are stated in Sec 3.3
- The notation should be optimized significantly. As the prime example, parsing notation like $\bar{\hat{\tilde{F}}}$ is quite challenging.
- The figures are low-quality, and their legends, axis labels, and axis ticks are completely unreadable.
- Sections 4.3.1 and 5 are extremely terse and poorly explained. Several concepts are not defined or referenced (e.g., spectral derivatives), and a reader like me, who's unfamiliar with dynamical systems, will get completely lost.
- The Related Works section is far too technical at the beginning of the paper. I suggest the authors move it to the end of the paper and also simplify its content.
- As a more minor point, please use `\sin` and `\mathrm{sinc}`; or ideally, define `\sinc` using `\DeclareMathOperator{\sinc}{\mathrm{sinc}}`.

Furthermore, I found the proofs of Prop 3.3, Thm 3.7, and Lemma 3.8 in Appendix B to be much too terse for a machine learning venue. For a reader unfamiliar with functional analysis, the proofs are completely inaccessible. Furthermore, the proofs that the translates of the activation functions studied form a weak Riesz basis, i.e., that they fulfill condition 1 in Def 3.1, are missing.

E.g., Prop 3.3 could be clarified significantly by stating and referencing the Poisson summation formula explicitly and stating that an equivalent condition for the integer translates of $F \in L_2(\mathbb{R})$ to form a Riesz basis is that $A \leq \sum_{k \in \mathbb{Z}} |\hat{F}(\omega + 2\pi k|^2 \leq B$ where $\hat{F}$ is the Fourier transform of $F$, which can be shown in a couple of lines.

Finally, I think the author's claim that $\mathrm{sinc}$ is optimal among INR activations is misleading because they do not state a clear optimality condition with respect to which the claim holds. What the authors mean is that $\mathrm{sinc}$ is the only activation among the ones they study for which their universal approximation theorem holds. I think the authors should clarify this in the text.

If the authors improve the writing by the end of the rebuttal period, I will be happy to recommend acceptance for the paper.

**Questions:**

- To define a notion of optimality of an activation function for a given class of functions, maybe the authors derive bounds on the growth of $n(\varepsilon)$ in Thms 3.4 and 3.10?
- It would be interesting to investigate whether gradient descent can recover the 2-layer INR weight settings that the authors use for their constructions in Thms 3.4 and 3.10.
- While the universal approximation theorems hold for 2-layer $\mathrm{sinc}$-INRs, it is unclear to me whether $\mathrm{sinc}$ retains its advantage in deeper architectures. Could the authors comment on this?

---

> ### Author Response · Authors · 2023-11-15
> **Response to reviewer REJk**
>
> We thank the reviewer for carefully going through our paper and providing suggestions to improve our current version. We have addressed all your comments. Please find detailed answers below. Please note that the Theorem and proposition numbers may have changed in the revised version as we have now moved related works to the end of the paper as you suggested.
>
> **There are countless typos in the paper, including undefined notation, which makes parsing the contents quite challenging. For example, I believe that in the paragraph below Eq 5, T is undefined, and I think it should be 1/Omega instead. Similarly, there's an inequality.....**
>
> Thank you! You are right $T$ is incorrect and it should actually be $\Omega$ not $1/\Omega$. The inequality in thm. 3.7 has also been corrected.
>
> **This is an even more serious problem in the appendix, where it appears that the content wasn't even re-read once, as there are gross typos (including some mathematical ones) in every second to third sentence....**
>
>
>
> 1. We have gotten rid of Appendix A as it was not needed in the main paper.
> 2. Appendix B is now referencing sec. 3.3 for the proofs correctly.
> 3. There was a typo with the notation $\overline{\hat{\widetilde{F}}}$. There should be no bar over this. We have fixed this.
>
>
> **The notation should be optimized significantly.....**
>
> We understand the reviewers sentiment with the notation. However, we would like to clarify this is the notation that is generally used in sampling theory employing functional analytic methods. Our reason for choosing this notation is to stick with convention. However there is a mistake with the notation in equation (6). There should be no bar on the Fourier transform of the test function. We have removed this.
>
> In general, the standard convention is
> $\widetilde{F}$ is used to denote a test function (in the sense of distributions) associated to $F$, and a $\hat{F}$ has always been used to denote the Fourier transform. Unfortunately, these conventions are normally fixed in the field of functional analysis. It happens to be the case that we need to use both these operations and hence that is why we had to go with that notation. Nonetheless, if the reviewer likes we are happy to replace the notation
> $\widehat{\widetilde{F}}$ with a completely different notation but this will then end up loosing how this function is connected to $F$.
>
> **Sections 4.3.1 and 5 are extremely terse and poorly explained. Several concepts are not defined or referenced (e.g., spectral derivatives), and a reader like me, who's unfamiliar with dynamical systems, will get completely lost.**
>
> Thank you for the suggestion. We have rewritten the above sections in more detail now. To facilitate the space constraints, we have moved some qualitative figures to appendix.
>
> **The Related Works section is far too technical at the beginning of the paper. I suggest the authors move it to the end of the paper and also simplify its content.**
>
> Thank you. We have addressed this and moved the related works to the end of the paper now.
>
> **As a more minor point, please use ...**
>
> We have changed $sin$ to $\sin$ and sinc to $\mathrm{sinc}$. Thank you for the suggestion.
>
> **Furthermore, I found the proofs of Prop 3.3, Thm 3.7, and Lemma 3.8 in Appendix B to be much too terse...**
>
> We presume you mean the proofs of Prop. 3.3, Thm. 3.4 and Th. 3.10. We greatly appreciate the reviewer taking the time to go through the proofs of these theorems. It is by no means easy as it uses deep mathematics. To clarify aspect of the proofs, we have added more details explaining several of the steps along the way and given clear references where needed. Hopefully now it is much more accessible. Thank you for pointing this out.
>
> **To define a notion of optimality of an activation function for a given class of functions, maybe the authors derive bounds on the growth of $n(\epsilon)$ in Thms 3.4 and 3.10?**
>
> By an activation being optimal we mean this in regards to whether it is a Riesz basis or not. This is certainly not a unique condition, there could be various different activations that form a Riesz basis but amongst the ones used by practitioners, namely, ReLU, tanh, sine, Gaussian, sinc, wavelets, we find that sinc is the only one that can generate a Riesz basis, making it an optimal activation to use from our theorem 3.10. We thank the reviewer for their comment, we have included a paragraph in Remark 2.11 (see revised version of our paper) clearly explaining this.
>
>  Understanding the growth rate of $n(\epsilon)$ is a very interesting problem, though at this point is out of the scope of this work. The growth of $n(\epsilon)$ will allow us to prove an effective version of theorem 3.10, allowing one to understand exactly how much overparameterization a network needs in order to fit such signals. We hope to carry out such a study in a future project.

---

> ### Author Response · Authors · 2023-11-15
> **Response to reviewer REJk - Part 2**
>
> **While the universal approximation theorems hold for 2-layer -INRs, it is unclear to me whether sinc retains its advantage in deeper architectures. Could the authors comment on this?**
>
> This is a very interesting question from the reviewer. What makes the situation for general depth difficult is that when you add depth you are composing activation functions. For example for a 3-layer network, employing $F$ as activation, the structure of the network becomes $W_3F(W_2F(W_1x + b_1) + b_2) + b_3$, where $W_i$, $b_i$ denote the weights and biases of each layer. In order to apply our theory to this situation, we would need to understand whether the composition of $F$ with itself can form a Riesz basis.
> In general, if $F$ forms a Riesz basis then there are no conditions guaranteeing that $F \circ F$ will satisfy the conditions of a Riesz basis. As far as we know there are no conditions guaranteeing that $F \circ F$ is a Riesz basis when $F$ is. Understanding how to generalize the theory to deep networks is something we are considering for future work.
>
> **It would be interesting to investigate whether gradient descent can recover the 2-layer INR weight settings that the authors use for their constructions in Thms 3.4 and 3.10.**
>
> This is also a very interesting question from the reviewer. Understanding whether gradient descent can actually converge to the weights that allow for thms. 3.4 and 3.10 to be valid is an important question, though we feel a theoretical proof showing this is out of scope for this work.
> We do thank the reviewer for bringing this up and we hope to take it up in a future project. Empirically it seems to be the case that gradient descent/Adam is converging to the optimum weights as we find from our experiments that sinc INRs are often giving the best PSNRs.

---

> ### Author Response · Authors · 2023-11-21
> **A kind reminder to the reviewer**
>
> Dear reviewer, we made the changes you had asked for and uploaded a revision of our paper on the 15th of November. We were wondering if you had the chance to review our changes and if you are happy with it. It would be great to get some feedback as to whether the changes we made were all done appropriately. We thank you for taking the time to review our paper and greatly appreciate your feedback.

---

> > ### Comment · Reviewer_REJk · 2023-11-21
> > **Response to the authors**
> >
> > I thank the authors for their elaborate response. I have skimmed the updated manuscript, and I am mostly satisfied with it; I have raised my score to reflect this. However, the Figures 1-3 in the main text are still unreadable. They are pixelated and the font size used for the tick and axis labels as well as the legend is too small. If the authors fix this, I will be happy to recommend acceptance.

---

> > > ### Author Response · Authors · 2023-11-22
> > > **Response to the reviewer**
> > >
> > > Dear reviewer,
> > >
> > > We have enhanced the figures you mentioned. We hope this revision will enable you to raise your score to acceptance. Thank you for the suggestions for improving our paper.

---

### Official Review · Reviewer_9zZJ · 2023-11-05

**Soundness:** 3 good
**Presentation:** 4 excellent
**Contribution:** 3 good
**Rating:** 6
**Confidence:** 4

**Summary:**

In this paper, the authors establish a connection between sampling theory and implicit neural representations, advocating for the sinc function as a better activation function with theoretical backing to minimize reconstruction error to any desired positive threshold. By demonstrating how shifted sinc functions constitute a Riesz basis—unlike common activation functions like sinusoidal and ReLU, which form weak Riesz bases—the paper provides a solid theoretical foundation for the sinc function's advantages. Finally, the authors showed the effectiveness of sinc functions on a variety of implicit representation tasks such as NeRFs, image reconstruction, and modeling dynamical systems.

**Strengths:**

**Coherent Narrative:** The paper presents a well-structured narrative, effectively linking various theorems to construct the main argument. The authors maintain a clear focus on their main message, presenting sufficient detail without overwhelming the reader, which facilitates a smooth reading experience and understanding of the core concepts.

**Novel Perspective on Implicit Representations:** The authors offer a fresh angle on implicit neural representations by examining them through the lens of sampling theory.

**Insights into Dynamical Systems:** The application of implicit neural representation perspectives to model dynamical systems is both novel and valuable and extends the utility of implicit neural representations.

**Weaknesses:**

**Limited Analysis of Failure Modes:** The paper's primary limitation is the insufficient exploration of the failure modes of the sinc activation function. Although the authors illustrate that a two-layer MLP can utilize sinc functions to approximate a signal within an acceptable error margin, this guarantee and the suggested architecture in B.1.2 is closely tied to a specific two-layer architecture. The emphasis on shallow MLPs might stem from this architectural dependency(Correct me if I am wrong). Understanding where the sinc activation function falls short or lacks expressiveness compared to other functions is a valuable addition to the paper.

**Clarification of Notation:** Regarding the mathematical notation, it would enhance clarity to emphasize the argument 'x' in Equation 3, written as $A_\Omega(s(x))$.

**Questions:**

**Q1:** How does increasing the depth of the neural network affect the theoretical guarantees associated with Riesz basis functions?
Could you elucidate on potential failure modes of the sinc activation function? Are there particular scenarios where traditional activation functions might outperform sinc?

**Q2:** Considering that state-of-the-art INRs often employ positional encoding (which by themselves can be deemed as an activation function), how does the sinc activation function interact with common positional encodings like random Fourier features[1] or sinusoidal encodings[2]? Moreover, is there a positional encoding strategy that effectively incorporates the sinc function?


[1] Tancik, Matthew, et al. "Fourier features let networks learn high frequency functions in low dimensional domains." Advances in Neural Information Processing Systems 33 (2020): 7537-7547.


[2] Vaswani, Ashish, et al. "Attention is all you need." Advances in neural information processing systems 30 (2017).

---

> ### Author Response · Authors · 2023-11-15
> **Response for the reviewer 9zZJ**
>
> We would like to thank the reviewer for their valuable comments for improving our paper. Please find our answers below.
>
> **Clarification of Notation: Regarding the mathematical notation, it would enhance clarity to emphasize the argument 'x' in Equation 3...**
>
> Thank you for the suggestion. We have made this change. Please see the revised version of our paper.
>
> **How does increasing the depth of the neural network affect the theoretical guarantees associated with Riesz basis functions? Could you elucidate on potential failure modes of the sinc activation function? Are there particular scenarios where traditional activation functions might outperform sinc?**
>
> This is a very interesting question from the reviewer. What makes the situation for general depth difficult is that when you add depth you are composing activation functions. For example for a 3-layer network, employing $F$ as activation, the structure of the network becomes $W_3F(W_2F(W_1x + b_1) + b_2) + b_3$, where $W_i$, $b_i$ denote the weights and biases of each layer. In order to apply our theory to this situation, we would need to understand whether the composition of $F$ with itself can form a Riesz basis.
> In general, if $F$ forms a Riesz basis then there are no conditions guaranteeing that $F \circ F$ will satisfy the conditions of a Riesz basis. As far as we know there are no conditions guaranteeing that $F \circ F$ is a Riesz basis when $F$ is. Understanding how to generalize the theory to deep networks is something we are considering for future work.
>
> Our theory implies the sinc activation is optimal for fitting signals in $L^2$. However, for more general function spaces such as $L^1$ the sinc activation would not be a good choice, as there is no $L^1$ sampling theorem for sinc. Thus for target signals that lie in $L^1$ but not $L^2$ sinc would not be guaranteed to provide a good choice for interpolation. We remark that almost all signals used in practise lie in $L^2$ so this is not a real disadvantage.
>
> In general, activations such as sinc are useful has they have a bandwidth hyperparameter that allows sinc networks to fit high frequency target signals. This is why they are able to overcome spectral bias. However, in contexts where one is trying to fit a very low frequency signal, such as image classification, sinc will not add much value and this is generally why you don't see activation such as sinc, Gaussian or sine being used in image classification.
>
> **Considering that state-of-the-art INRs often employ positional encoding (which by themselves can be deemed as an activation function), how does the sinc activation function interact with common positional encodings like random Fourier features[1] or sinusoidal encodings[2]?**
>
> We have compared image reconstruction performance against RFF positional encodings in Figure 1. As shown, sinc activations outperform positional encodings by a significant margin. It has also shown in [1] and [2] that Gaussian and Wavelet activations also outperform RFF positional encoding despite the fact that practitioners often tend to use RFF encodings in INRs out of habit, although strong empirical evidence suggests against it. We tried using sinc activations in conjunction with the RFF positional embeddings, but observed that this setting deteriorates the performance of the RFF embeddings.
>
> [1] - Ramasinghe, Sameera, and Simon Lucey. "Beyond periodicity: Towards a unifying framework for activations in coordinate-mlps." European Conference on Computer Vision. Cham: Springer Nature Switzerland, 2022.
>
> [2] - Saragadam, Vishwanath, et al. "Wire: Wavelet implicit neural representations." Proceedings of the IEEE/CVF Conference on Computer Vision and Pattern Recognition. 2023.

---

> ### Author Response · Authors · 2023-11-15
> **Response for the reviewer 9zZJ - Part 2**
>
> **Moreover, is there a positional encoding strategy that effectively incorporates the sinc function?**
>
> This is an excellent question. Recent research, notably [3], has provided compelling evidence that the effectiveness of positional encodings need not be exclusively tied to a Fourier perspective. They demonstrate that non-Fourier embedding functions, such as shifted Gaussian functions, can be effectively utilized for positional encoding. These functions are characterized by having a sufficiently high Lipschitz constant and the ability to generate high-rank embedding matrices, attributes that are shown to achieve results comparable to Random Fourier Feature (RFF) encodings.
>
> Building on this, research in [4] further confirms that shifted Gaussian functions with spatially varying variances can surpass the performance of RFF encodings. Given that sinc functions also exhibit these desirable properties, they can be feasibly employed as shifted basis functions for high-frequency signal encoding.
>
>
> To explore this, we developed a sinc-based positional embedding layer. For a 2D coordinate $(x_1,x_2)$, each dimension is embedded using sinc functions:
>
> $$
> \psi_1(x_1) = [\mathrm{sinc}(\|t_1 - x_1\|), \mathrm{sinc}(\|t_2 - x_1\|), \dots, \mathrm{sinc}(\|t_N - x_1\|),
> $$
>
> $$
> \psi_1(x_2) = [\mathrm{sinc}(\|t_1 - x_2\|), \mathrm{sinc}(\|t_2 - x_2\|), \dots, \mathrm{sinc}(\|t_N - x_2\|),
> $$
> where $t_1, \dots, t_N$ are equidistant samples in $[0,1]$. Then, these embeddings are concatenated to create the final embedding as,
>
> $$
> \Psi(x_1,x_2) = [\psi_1(x_1), \psi_1(x_2)]
> $$
>
> In a comparative study using the DIV2K dataset for image reconstruction, our sinc-based positional embedding layer demonstrated superior performance to an RFF-based layer, as shown in the table below. The shown metrics are averaged values over the dataset:
>
> | PE layer | PSNR |
> |----------|------|
> | RFF      | 23.5 |
> | Sinc PE  | 26.4 |
>
> This result indicates that sinc-based positional embeddings offer a promising alternative to RFF encodings. We have included a short discussion of sinc positional embeddings in the supplementary material and anticipate this will stimulate further research in this area.
>
>
> [3] - Zheng, Jianqiao, Sameera Ramasinghe, and Simon Lucey. "Rethinking positional encoding." arXiv preprint arXiv:2107.02561 (2021).
>
> [4] - Ramasinghe, Sameera, and Simon Lucey. "A learnable radial basis positional embedding for coordinate-MLPs." Proceedings of the AAAI Conference on Artificial Intelligence. Vol. 37. No. 2. 2023.

---

> > ### Comment · Reviewer_9zZJ · 2023-11-22
> > **Response to Authors**
> >
> > I have read the authors' comments and I thank the authors for addressing my concerns. I believe the explanations were convincing and now I have a better understanding of the theoretical guarantee of the shifted sinc functions. Moreover, I agree  that the results corresponding to the sinc positional encoding look interesting and require further investigations in future works.

---

### Meta-Review · Area_Chair_iMTY · 2023-12-12

**Metareview:**

Summary: The article investigates activations in implicit neural representations in regard to universal approximation and identifies the sinc activation as a good choice.

Strengths: Referees find the article is well structured and consider the presented links between implicit representations and sampling theory to obtain theoretical results on universal approximation depending on activation functions novel, simple, effective and interesting.

Weaknesses: At the same time, there were concerns about the quality of writing, lack of intuitions, limited analysis of failure modes, theoretical guarantees for deep networks, and necessary conditions. There was a good amount of discussion during the discussion period. Some concerns could be clarified and some reviewers updated their scores accordingly.

At the end of the discussion period, the referees regard the article as borderline. After considering the discussion and taking a look a the article, I find the article obtains promising results but that there are various aspects that if investigated more thoroughly could further strengthen the work, particularly considering the weaknesses listed above. Thus, in consideration of a very high bar of acceptance for this conference, I must reject the article at this time.

**Justification For Why Not Higher Score:**

Although the work presents valuable results, a more thorough development of implications, necessary conditions, applicability to deep setting, positional encodings could still strengthen the contribution.

**Justification For Why Not Lower Score:**

NA

---

### Decision · Program_Chairs · 2024-01-16

Reject